# DIVERSITY-INCENTIVIZED EXPLORATION FOR VERSATILE REASONING

**Zican Hu[12]∗**    **Shilin Zhang[12]∗**    **Yafu Li[24]†✉**    **Jianhao Yan[23]**    **Xuyang Hu[2]**
**Leyang Cui[3]**    **Xiaoye Qu[2]**    **Chunlin Chen[1]**    **Yu Cheng[4]✉**    **Zhi Wang[12]✉**
[1] Nanjing University [2] Shanghai AI Laboratory
[3] Westlake University [4] The Chinese University of Hong Kong
{zicanhu,shilinzhang}@smail.nju.edu.cn   yafuly@gmail.com
chengyu@cse.cuhk.edu.hk   zhiwang@nju.edu.cn

## ABSTRACT

Reinforcement Learning with Verifiable Rewards (RLVR) has emerged as a crucial paradigm for incentivizing reasoning capabilities in Large Language Models (LLMs). Due to vast state-action spaces and reward sparsity in reasoning tasks, existing methods often struggle with deficient exploration and poor sample efficiency. In the paper, we propose **DIVER** (**D**iversity-**I**ncentivized Exploration for **V**ersatil**E R**easoning), an innovative framework that highlights the pivotal role of global sequence-level diversity to incentivize deep exploration for versatile reasoning. We first conduct a primary empirical study to reveal a strong positive correlation between global diversity and reasoning capacity. Building on this insight, we introduce global diversity incentives as an intrinsic reward to promote deep exploration in a semantically structured space. Incorporating the intrinsic reward, we develop a potential-based reward shaping mechanism to preserve optimal policy invariance and design simple heuristics to mitigate possible reward hacking. Experimental results show that DIVER outperforms competitive RLVR baselines with various exploration strategies on both in-domain and out-of-domain tasks, excelling in both Pass@1 and Pass@k evaluations. Our code is available at https://github.com/NJU-RL/DIVER.

## 1 INTRODUCTION

Reinforcement Learning with Verifiable Rewards (RLVR) has advanced reasoning capabilities in Large Language Models (LLMs) through rule-based verification on model's responses (Guo et al., 2025; Hu et al., 2025a; Zeng et al., 2025). A central challenge is the fundamental exploration–exploitation tradeoff highlighted in classic RL literature (Lillicrap et al., 2015; Haarnoja et al., 2018). Unlike traditional RL environments with relatively small, well-defined state-action spaces (Sutton & Barto, 2018; Wang et al., 2024b; Zhang et al., 2025), LLM policies operate in vast, high-dimensional textual spaces with complex semantics, where the number of possible state-action pairs grows

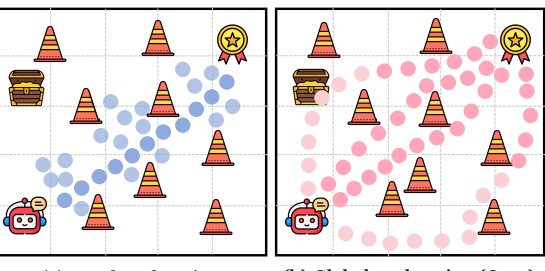

(a) Local exploration    (b) Global exploration (Ours)

Figure 1: Local token-level vs. Global sequence-level exploration. We incentivize deep exploration to broaden diverse pathways for versatile reasoning.

exponentially with sequence length (Gupta et al., 2024; Ahn et al., 2024; Hu et al., 2025b). This combinatorial explosion greatly increases the difficulty of effective exploration in textual reasoning, especially under limited computing resources. Furthermore, the inherent reward sparsity in

---

∗Equal contributions. Zican Hu and Shilin Zhang are listed alphabetically by last name. This work was conducted during internship at Shanghai AI Laboratory. ✉Corresponding authors. †Project lead.

challenging reasoning tasks creates massive "reward deserts" where the model receives little to no meaningful feedback most of the time (Haarnoja et al., 2017; Wu et al., 2025), hindering the discovery of improved pathways due to the lack of valid feedback signals (Zheng et al., 2024; Zhan et al., 2026). Therefore, existing methods often struggle with deficient exploration and poor sample efficiency (Deng et al., 2025; Chen et al., 2025b).

Thus, introducing efficient exploration mechanisms becomes essential for enabling LLMs to continually self-improve their reasoning abilities through the iterative trial-and-error cycle (Ladosz et al., 2022; Song et al., 2025). Current LLM literature has explored several attempts based on increasing the uncertainty in the policy's action selection process (Yao et al., 2025), such as alleviating policy entropy collapse (Cui et al., 2025b), branching only high-entropy tokens (Liu et al., 2025a), and retaining policy gradient updates for high-entropy tokens only (Wang et al., 2025). These dithering strategies inject randomness into the policy's action distribution, facilitating exploration by promoting the selection of uncertain actions. Typically, they incorporate diversity at the *local* action level (i.e., token level) to help the policy escape local optima and accelerate training.

While action-level uncertainty estimates allow the agent to direct its exploration toward potentially informative states, they are insufficient on their own to ensure efficient exploration (Osband et al., 2016). Provably efficient schemes require far-sighted, *deep* exploration that is directed over multiple timesteps, inducing temporally-extended diversity at a higher level (Osband et al., 2019). Unlike local exploration, deep exploration is essential to significantly broaden the *global* sequence-level diversity of reasoning pathways, stimulating the model to discover novel and effective solution patterns. This principle of optimizing global diversity is essential for advancing the deep exploration capabilities of frontier RL algorithms (Eysenbach et al., 2019; Grillotti et al., 2024), showcasing remarkable efficiency for solving intricate tasks in a more human-like manner (Celik et al., 2024). However, efficient deep exploration mechanisms remain largely underexplored in LLM reasoning.

Built on these insights, we propose **DIVER** (**D**iversity-**I**ncentivized Exploration for **V**ersatil**E** **R**easoning) that emphasizes the pivotal role of global sequence-level diversity to incentivize deep exploration for versatile LLM reasoning. We first conduct a primary empirical study where evidence reveals a strong positive correlation between global diversity and reasoning capacity. This finding motivates us to explicitly optimize sequence-level diversity during RL training. To this end, we formulate the global diversity across group responses as an intrinsic reward, incentivizing deep exploration in a semantically structured space. When incorporating this intrinsic reward, we design a potential-based reward shaping mechanism to preserve optimal policy invariance and develop simple heuristics to mitigate possible reward hacking. Specifically, we employ two easy-to-implement metrics to quantify the diversity inherent across group responses, Textual Diversity and Equational Diversity, while in principle any other metrics are compatible with our framework. We hope this study inspires further investigation into global diversity and incentivizes efficient deep exploration mechanisms for broadening LLM's versatile reasoning capacities.

Experimental results demonstrate that DIVER consistently outperforms competitive RLVR baselines across six math reasoning benchmarks (AIME24/25, AMC, OlympiadBench, Minerva, MATH500). Notably, DIVER shows stronger generalization capabilities with a **+3.2** points improvement over the GRPO baseline on out-of-domain benchmarks (ARC-c, GPQA*, MMLU-Pro). To evaluate exploration effectiveness through multi-attempts, we employ the Pass@k metric, where DIVER consistently surpasses all baselines. The most substantial gain appears on AIME25, where DIVER achieves a **+6.7** points improvement in Pass@32 performance. Our in-depth analysis reveals that DIVER's advantage stems from its global sequence-level diversity and deep exploration capability.

## 2 RELATED WORK

**Exploration in RL.** Exploration techniques are key to solving high-dimensional, sparse-reward RL problems (Ladosz et al., 2022). They can be roughly categorized into three kinds: 1) injecting stochastic noise into behavior policies (Lillicrap et al., 2015; Fujimoto et al., 2018); 2) incorporating policy entropy into the optimization objective (Haarnoja et al., 2017; 2018); and 3) introducing intrinsic rewards independent of environmental feedback, such as count-based bonuses (Bellemare et al., 2016; Wang et al., 2026), information gains (Houthooft et al., 2016), or the novelty of experience (Pathak et al., 2017; Burda et al., 2019). Our work extends the third category from classical RL to LLM reasoning tasks. Frontier RL algorithms adopt the principle of promoting global diversity

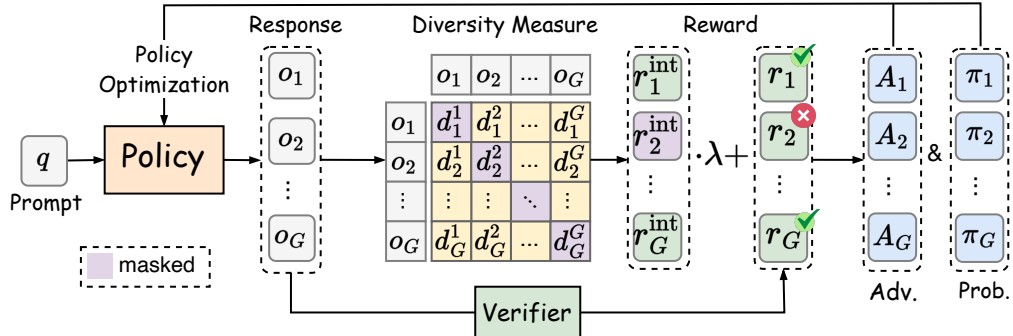

Figure 2: Overview of DIVER where we formulate the global sequence-level diversity of response $o_i$ within a group of $G$ rollouts as an intrinsic reward $r_i^{\text{int}}$ to incentivize deep exploration. Diversity incentives are applied to correct solutions only to align shaping rewards with the true objective.

to improve their deep exploration capabilities (Liu et al., 2026), such as learning diverse skills in an unsupervised manner (Eysenbach et al., 2019), inducing diverse behaviors with constrained optimization (Grillotti et al., 2024), and acquiring diverse skills using mixture-of-experts (Celik et al., 2024). As efficient deep exploration mechanisms remain largely underexplored in LLM reasoning, our method aims to address this gap.

**Diversity for LRMs.** Current literature has attempted to promote diversity in LLM reasoning by increasing the uncertainty in the action selection process. Some studies directly manage individual tokens at a micro level. (Cui et al., 2025b) applies clip and KL penalty constraints to tokens that tend to cause entropy collapse. (Liu et al., 2025a) selectively branches high-entropy tokens only to enhance exploration in test-time RL. (Wang et al., 2025) leverages high-entropy minority tokens to steer the model toward diverse reasoning pathways. Other approaches achieve similar results by reformulating optimization objectives. (Yao et al., 2025) injects a token-level diversity measure into policy optimization. (Cheng et al., 2025) augments the advantage function with an entropy-based term. (Yu et al., 2025) increases the upper bound for clipping the importance sampling ratio to emphasize low-probability tokens. In summary, these methods typically incorporate diversity at the local action level, facilitating exploration by promoting the selection of uncertain actions.

Recently, (Chen et al., 2025b) uses the Pass@k metric as the training reward, leveraging multiple candidate solutions in one trial to enhance the exploration abilities of LLMs. However, it does not explicitly account for global diversity, since it does not seek to optimize diversity across candidate solutions. A concurrent work is (Li et al., 2025) that trains a partitioning classifier to measure diversity and amplifies the advantage function by the diversity assessment. Beyond algorithmic-level exploration strategies, data-level approaches have also been proposed to improve training efficiency. Prompt selection methods (Chen et al., 2025a; Bae et al., 2025; Qu et al., 2025) filter training samples based on difficulty or informativeness to enhance exploration efficiency. The key distinction of our method lies in how diversity is measured and how it is embedded within policy optimization.

## 3 METHOD

In this section, we first give the problem statement where the reasoning task is formulated as an RL problem. Then, we present a primary empirical study to show the impact of global sequence-level diversity on reasoning performance. Finally, we introduce DIVER in detail, with principled formulations to quantify diversity, guarantee optimal policy invariance, and mitigate reward hacking.

### 3.1 PROBLEM STATEMENT

RL is based on the Markov decision process (MDP) formulation with a tuple $(S, A, T, R, \gamma)$, where $S/A$ is the state/action space, $T(s'|s, a) : S \times A \times S \mapsto [0, \infty]$ is the transition operator that defines the probability density function of transitioning to state $s' \in S$ conditioned on taking action $a \in A$ in state $s \in S$, $R(s, a) : S \times A \mapsto \mathbb{R}$ is the reward function, and $\gamma \in (0, 1]$ is the discount factor.

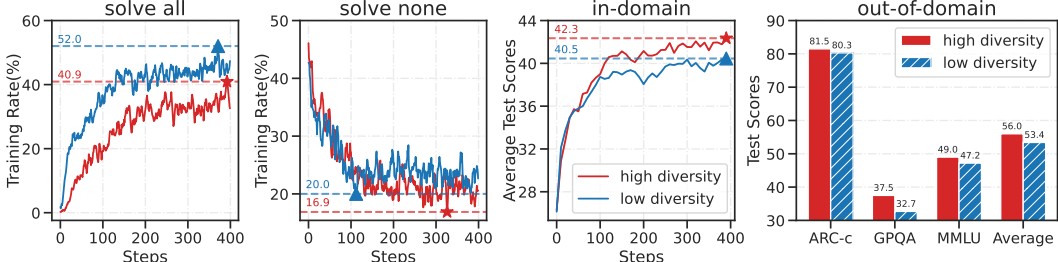

Figure 3: Performance comparison between high-diversity (red) and low-diversity (blue) training. `solve all`: Number of samples with all rollouts correctly solved. `solve none`: Samples with no correct rollouts. `in-domain`: Average test scores across training steps for in-domain benchmarks. `out-of-domain`: Final performance for out-of-domain benchmarks.

**Group Relative Policy Optimization (GRPO).** We build the DIVER framework upon GRPO (Shao et al., 2024), a prominent RLVR baseline that incentivizes reasoning capability in LLMs using a rule-based verifiable reward function. GRPO discards the critic model and instead computes advantages using rule-based rewards from group-level comparisons. For each query $q$, the policy $\pi_{\theta_{old}}$ generates $G$ candidate responses $\{o_1, ..., o_G\}$. Each response is evaluated by a binary reward function $r_i \in \{0, 1\}$ that checks whether the extracted answer matches the golden answer, yielding rewards $\{r_1, ..., r_G\}$. This verifiable reward design effectively mitigates reward hacking (Gao et al., 2023), enabling robust scaling of RL training. The policy $\pi_\theta$ is then updated by maximizing:

$$J_{\text{GRPO}}(\theta) = \mathbb{E}_{q \sim \mathcal{D}, \{o_i\}_{i=1}^G \sim \pi_{\theta_{old}}(\cdot|q)} \left[ \frac{1}{G} \sum_{i=1}^{G} \sum_{t=1}^{|o_i|} \text{CLIP}(\rho_{i,t}, A_i) - \beta KL(\pi_\theta || \pi_{\text{ref}}) \right], \quad (1)$$

where $\rho_{i,t} = \frac{\pi_\theta(o_{i,t}|q, o_{i,<t})}{\pi_{\theta_{old}}(o_{i,t}|q, o_{i,<t})}$ is the importance sampling ratio for the token at timestep $t$ in response $o_i$. The clipped objective $\text{CLIP}(\rho_{i,t}, A_i) = \min(\rho_{i,t}A_i, \text{clip}(\rho_{i,t}, 1 - \epsilon, 1 + \epsilon)A_i)$ ensures stable updates within the trust region (Schulman et al., 2017). The KL regularization term constrains the policy from deviating too far from a reference model $\pi_{\text{ref}}$. The advantage $A_i$ is computed as:

$$A_i = \frac{r_i - \mu_r}{\sigma_r}, \quad \text{where} \quad \mu_r = \frac{1}{G} \sum_{j=1}^{G} r_j, \quad \sigma_r = \sqrt{\frac{1}{G} \sum_{j=1}^{G} (r_j - \mu_r)^2}. \quad (2)$$

### 3.2 THE IMPACT OF SEQUENCE-LEVEL DIVERSITY ON RLVR

While a line of recent work (Liu et al., 2025a; Wang et al., 2025) focuses on promoting local token-level diversity, we explore and analyze how global sequence-level diversity within data affects RL fine-tuning for LRMs. We first conduct an illustrative experiment that compares training the model using rollouts with different levels of sequence-level diversity.

Figure 3 shows the performance of filtering GRPO rollouts into high-diversity (red) and low-diversity (blue) subsets to train separate models, using the proposed diversity metrics (TD and ED in Sec. 3.3). The experimental details are provided in the Appendix B.1. During training, high-diversity training simultaneously yields a lower "solve all" rate (1.1 points decrease) and a lower "solve none" rate (3.1 points decrease). Intuitively, the high-diversity scheme trades off a small fraction of responses to explore a wider range of reasoning patterns, enabling the search for novel breakthroughs when conventional solutions fail. This property is particularly appealing in practice, since achieving success on complex problems carries greater value than obtaining uniformly correct answers on simple questions. During testing, high-diversity training achieves higher test scores, with an increase of **+1.8** points for in-domain benchmarks and an increase of **+2.6** points for out-of-domain benchmarks. This encouraging result verifies that promoting global diversity can broaden LLM's reasoning capacity. A noteworthy point is that the superiority of high-diversity training is more pronounced for out-of-domain tasks compared to in-domain ones. This observation is consistent with our motivation, where emphasizing diversity enables exploring a broader spectrum of reasoning patterns and expands generalization capacity.

### 3.3 METRICS FOR QUANTIFYING SEQUENCE-LEVEL DIVERSITY

The above insight highlights the necessity of promoting global diversity to enable more effective exploration for RLVR, thereby incentivizing versatile LLM reasoning. Naturally, it is crucial to develop efficient metrics capable of quantifying sequence-level diversity in reasoning tasks. To this end, we design two easy-to-implement metrics, Textual Diversity and Equational Diversity. As a general framework, our method is also compatible with any other diversity metrics.

**Textual Diversity (TD).** It refers to the text-level mutual dissimilarity across group responses from a given query. We adopt the BLEU score (Papineni et al., 2002), a popular automated and inexpensive metric that correlates highly with human evaluation, to quantify the text similarity, and invert it to measure dissimilarity. In essence, BLEU measures similarity by calculating the overlap of n-grams (word sequences) between a candidate text and the reference text, with higher scores indicating greater similarity. Appendix C.2 presents the detailed definition.

For each candidate response $o_i$ in the group $\{o_1, ..., o_G\}$, we calculate its BLEU similarity to every other response and derive its TD within the group as

$$\text{TD}(o_i) = \frac{1}{G-1} \sum\nolimits_{j \in [G] \setminus \{i\}} \Big(1 - \text{BLEU}(o_i, o_j)\Big). \tag{3}$$

**Equational Diversity (ED).** It refers to the differences in the formulas contained across group responses to a given query, aiming to characterize underlying reasoning patterns for mathematical tasks. A math problem often admits multiple solutions with distinct formulaic patterns, and exploring diverse problem-solving approaches can increase the likelihood of finding the correct answer.

Let $\mathcal{F}(o_i)$ denote the set of formulas extracted from response $o_i$, and $\mathcal{F}_{-i}$ denote the set of formulas collected from other responses in the same group as

$$\mathcal{F}_{-i} = \bigcup\nolimits_{j \in [G] \setminus \{i\}} \mathcal{F}(o_j), \quad [G] = \{1, \ldots, G\}. \tag{4}$$

Then, the per-response equational diversity is defined as the ratio of unique formulas in response $o_i$ to the total number of formulas it contains:

$$\text{ED}(o_i) = \frac{\big|\mathcal{F}(o_i) \setminus \mathcal{F}_{-i}\big|}{\big|\mathcal{F}(o_i)\big|}, \quad \text{if } |\mathcal{F}(o_i)| > 0; \quad \text{or } 0, \text{ otherwise.} \tag{5}$$

### 3.4 PROMOTING GLOBAL DIVERSITY FOR DEEP EXPLORATION

The findings in Sec. 3.2 motivate us to explicitly optimize global diversity during RL training. As shown in Figure 2, we formulate the global diversity as an intrinsic reward to incentivize deep exploration in a semantically structured space. For a group of responses $\{o_1, ..., o_G\}$, we calculate the pairwise dissimilarity using the proposed diversity metrics in Sec. 3.3, yielding a $G \times G$ matrix $D$ where each element $d_i^j$ denotes the diversity between responses $o_i$ and $o_j$. Then, the diversity of response $o_i$ within the group is calculated by simply averaging its dissimilarities to all others as $d(o_i) = \frac{1}{G-1} \sum_{j \neq i}^{G} d_i^j$, i.e., averaging across the corresponding row in the diversity matrix $D$.

To promote global diversity in RLVR, a natural option is to directly supply the quantified diversity as an additional reward to guide the learning process. However, this naive shaping can change the optimal policy and mislead the agent into learning suboptimal policies (Ng et al., 1999). Hence, we adopt a potential-based reward shaping scheme to preserve optimal policy invariance when incorporating the intrinsic reward (Wang et al., 2023; Müller & Kudenko, 2025). We formulate the **intrinsic reward** $R_{\text{int}}$ as the difference between the sequence-level diversities $d(\cdot)$ of adjacent states as

$$R_{\text{int}}(s_t, a_t, s_{t+1}) = \gamma d(s_{t+1}) - d(s_t), \tag{6}$$

where $d(\cdot)$ is the exactly the potential function over states $s \in S$. In the LLM setting of Eq. (1), the states and action within a given query-response pair are defined as $s_t := [q, o_{i, \leq t}]$, $a_t := o_{i,t+1}$, and $s_{t+1} := [q, o_{i, \leq t+1}]$. Then, the specific intrinsic reward becomes

$$R_{\text{int}}([q, o_{i, \leq t}], o_{i,t+1}, [q, o_{i, \leq t+1}]) = \gamma d([q, o_{i, \leq t}]) - d([q, o_{i, \leq t+1}]), \quad i = 1, ..., G. \tag{7}$$

Since GRPO inherits the PPO principle (Schulman et al., 2017) that derives policy gradients at the sequence level, the intrinsic reward for a complete query-response pair is calculated as

$$
\begin{aligned}
R_{\mathrm{int}}\Big([q, o_i]\Big) &= \sum_{t=0}^{T-1} \gamma^t R_{\mathrm{int}}\Big([q, o_{i,\leq t}], o_{i,t+1}, [q, o_{i,\leq t+1}]\Big) \\
&= \sum_{t=0}^{T-1} \gamma^t [\gamma d([q, o_{i,\leq t+1}]) - d([q, o_{i,\leq t}])] \\
&= \gamma^T d([q, o_{i,\leq T}]) - d(q) \\
&= \gamma^T d([q, o_i]),
\end{aligned}
\tag{8}
$$

where $T$ denotes the terminal step, and $d([q, o_i])$ is the global diversity of response $o_i$, which can be calculated by metrics in Sec. 3.3. The diversity of a constant query $q$ is zero, i.e., $d(q) = 0$.

This diversity-incentivized intrinsic reward complements the traditional rule-based assessment reward $R(\cdot)$. By incorporating the diversity measure, we design a versatile evaluation system that values both correctness and solution diversity, yielding the new reward function $R'(\cdot)$ as

$$
R'([q, o_i]) = R([q, o_i]) + \lambda \cdot R_{\mathrm{int}}([q, o_i]),
\tag{9}
$$

where $\lambda$ is the shaping ratio that balances between accuracy and diversity. Maximizing the intrinsic reward incentivizes the model to explore diverse reasoning pathways at the sequence level, facilitating the discovery of novel and effective solutions to complex problems. Finally, we substitute the augmented reward function $r_i' = R'([q, o_i])$ for the original reward $r_i = R([q, o_i])$ in Eq. (2) to calculate the advantage function under the GRPO framework.

When including the intrinsic reward, we will transform the original MDP $M = (S, A, T, R, \gamma)$ to a new one $M' = (S, A, T, R', \gamma)$, where $R' = R + \lambda R_{\mathrm{int}}$. Since we are learning a policy for the transformed MDP $M'$ in the hope of using it in the original one $M$, it is essential to ensure that this transformation does not mislead the agent into learning suboptimal policies. Theorem 1 guarantees the optimal policy invariance when incorporating global diversity as an intrinsic reward, validating the effectiveness of our reward shaping mechanism. Appendix A presents the detailed proof.

**Theorem 1** (Optimal Policy Invariance). *Let $M = (S, A, T, R, \gamma)$ denote the MDP for the LLM reasoning task. $d(\cdot) \colon S \mapsto \mathbb{R}$ is a real-valued function that computes the sequence-level diversity $d(s)$ of the state $s$ within a group of rollouts. We formulate $R_{int}(\cdot) \colon S \times A \times S \mapsto \mathbb{R}$ as an intrinsic reward function that is the difference between sentence diversities of two adjacent states, such that for all $s \in S, a \in A, s' \in S$, $R_{int}(s, a, s') = \gamma d(s') - d(s)$. Then, with any constant balancing ratio $\lambda$, every optimal policy in the transformed MDP $M' = (S, A, T, R + \lambda R_{int}, \gamma)$ will also be an optimal policy in $M$, and vice versa.*

The ingenuity of our reward-shaping design lies in that by setting the shaping reward as the difference between the diversities of adjacent states as in Eq. (6), the intrinsic reward for a complete query-response pair is derived as the diversity of the final response as in Eq. (8). This elegant formulation avoids the need to calculate the diversity of any intermediate sentences, which saves a significant amount of computation while ensuring optimal policy invariance.

## 3.5 Mitigating Reward Hacking

Including an additional shaping reward could increase the risk of reward hacking, a phenomenon where an RL agent exploits flaws or ambiguities in the reward function to achieve high rewards without genuinely solving the intended task (Pan et al., 2022). This is particularly concerning in language models, where the complex nature of reasoning tasks makes reward functions susceptible to biased exploitation (Liu et al., 2025b). Although DIVER preserves optimal policy invariance after reward shaping, the model may still over-exploit intrinsic rewards and neglect the primary objective during training. Since the primary reward for reasoning correctness is sparse and difficult to attain, obtaining the reward for diversity is considerably easier, especially when addressing hard problems. To this end, we design simple heuristics to mitigate the potential risk of reward hacking as follows. Ablation study in Sec. 4.4 verifies the successful mitigation of possible reward hacking.

**Balanced Shaping.** We clip the diversity reward to be $r_i^{\mathrm{int}} = \mathrm{clip}(r_i^{\mathrm{int}}; 0, \sigma)$, where $r_i^{\mathrm{int}} = R_{\mathrm{int}}([q, o_i])$ and $\sigma$ is a predetermined upper bound that prevents the model from excessively exploiting the shaping reward. Moreover, we gradually reduce the balancing ratio $\lambda$ during training.

Akin to the classic exploration-exploitation tradeoff in RL philosophy (Sutton & Barto, 2018), we prefer exploring diverse solutions early and tend to exploit accumulated knowledge later.

**Conditional Shaping.** We only include the shaping reward to correct responses within the group as $r_i' = r_i + \lambda \cdot r_{\text{int}}^i \cdot I(r_i)$, where $I(r_i)$ is an indicator function that equals 1 if the response is correct and 0 otherwise. This conditional shaping ensures that the diversity incentive only rewards genuinely correct solutions, preventing the model from trading off correctness for diversity. The design principle effectively aligns shaping rewards with the true objective, addressing potential reward hacking concerns while promoting valuable, diversified exploration across the solution space.

## 4 Experiments

We comprehensively evaluate and analyze our method to answer the following research questions: i) Can DIVER improve performance while maintaining effective global exploration and reliably extending to other models? ii) Can DIVER achieve an effective and broader exploration scope that unlocks enhanced reasoning capacity? iii) What is the appropriate configuration of DIVER for balancing diversity, reward stability, and exploration horizons?

### 4.1 Experimental Settings

**Datasets and Evaluation.** Our training data is a subset of OpenR1-Math-220k (Face, 2025), with prompts collected from NuminaMath 1.5 (LI et al., 2024) following the LUFFY (Yan et al., 2025). We evaluate on six mathematical reasoning benchmarks: AIME24/25, AMC (Li et al., 2024), Minerva (Lewkowycz et al., 2022), OlympiadBench (He et al., 2024), and MATH-500 (Hendrycks et al., 2021). Main results report Avg@32 for the smaller test sets (AIME24/25, AMC), and Pass@1 for others. For cross-domain, we test on ARC-c (Clark et al., 2018), GPQA-diamond (GPQA*) (Rein et al., 2024), and MMLU-Pro (Wang et al., 2024a).

**Baselines and Training.** We compare DIVER against two categories of baselines: 1) Established RLVR methods: `SimpleRL-Zoo`(Zeng et al., 2025), `OpenReasoner-Zero`(Hu et al., 2025a), and `PRIME-Zero` (Cui et al., 2025a). 2) Our reproduction of exploration RL methods: `GRPO w/ Clip-higher`(Yu et al., 2025), which modifies clip ratio to encourage exploration; `Entropy-RL`(Cui et al., 2025b), which addresses policy entropy collapse through covariance-based techniques; and `Pass@k Training` (Chen et al., 2025b), which uses Pass@k as the reward to adaptively balance exploration and exploitation. We set $\beta = 0$ to remove the KL loss term and use 0.28 for higher clip following GPPO w/ Clip-higher. Sample batch size is 128, update batch size is 32, with 8 rollouts per prompt. All experimental details are documented in Appendix B.2.

### 4.2 Main Results

**Reasoning Performance on Qwen2.5-Math-7B.** Table 1 illustrates DIVER's evaluation results compared to established RLVR methods (SimpleRL-Zoo, OpenReasoner-Zero, and PRIME-Zero). All implementations based on Qwen2.5-Math-7B. DIVER with Textual Diversity (TD) and Equational Diversity (ED) achieves average scores of **42.3** and **43.0** on six mathematical benchmarks, outperforming OpenReasoner-Zero by **+2.0** points. On challenging out-of-domain tasks, DIVER reaches **58.4** average score, surpassing OpenReasoner-Zero by **+6.8** points, with notable gains on ARC-c (**+10.1**) and GPQA (**+12.5**). These results confirm that encouraging diverse reasoning paths at the sequence level enhances model generalization significantly.

**Comparison with Exploration RL Methods.** We evaluate DIVER against representative exploration RLVR methods: GPRO w/ Clip-higher (*undirected exploration*), Entropy-RL (*action-level exploration*), and Pass@k Training (*within-group bootstrap sampling*). Experimental results demonstrate that DIVER outperforms the best exploration method, Entropy-RL, by **+1.2** points on average across in-domain tasks. This improvement is particularly pronounced on challenging benchmarks, with a **+4.6** point advantage on OlympiadBench. Notably, on out-of-domain tasks, both DIVER and the global exploration approach Pass@k Training exhibit superior generalization compared to local exploration techniques. DIVER surpasses the best local exploration method, Entropy-RL, by **+2.4**

Table 1: Performance comparison across in-domain and out-of-domain tasks based on `Qwen2.5-Math-7B`. Best results in **bold** and second best underlined. DIVER-TD and DIVER-ED represent our approach implemented with Textual Diversity and Equational Diversity, respectively. DIVER-MIX combines both metrics during training to achieve optimal diversity.

| Model | In-Domain Performance | | | | | | Out-of-Domain Performance | | | |
| --- | --- | --- | --- | --- | --- | --- | --- | --- | --- | --- |
| | AIME 24/25 | AMC | MATH-500 | Minerva | Olympiad | Avg. | ARC-c | GPQA* | MMLU-Pro | Avg. |
| Qwen2.5-Math-7B | 11.8/6.3 | 43.1 | 56.8 | 16.9 | 25.4 | 26.7 | 38.1 | 12.2 | 31.5 | 27.3 |
| *Previous RLVR methods* | | | | | | | | | | |
| SimpleRL-Zoo | **25.2**/12.0 | 57.6 | 76.2 | 27.2 | 41.0 | 39.9 | 22.0 | 20.4 | 32.5 | 25.0 |
| OpenReasoner-Zero | 16.5/15.0 | 52.1 | 82.4 | 33.1 | **47.1** | 41.0 | 66.2 | 29.8 | **58.7** | 51.6 |
| PRIME-Zero | 17.0/12.8 | 54.0 | 81.4 | **39.0** | 40.3 | 40.7 | 73.3 | 18.2 | 32.7 | 41.4 |
| *Exploration RL Methods* | | | | | | | | | | |
| GRPO w/ Clip-higher | 18.9/16.4 | 57.3 | 81.2 | 28.7 | 41.5 | 40.7 | 82.1 | 36.2 | 47.2 | 55.2 |
| Entropy-RL | 23.6/12.8 | 58.4 | 82.8 | 31.6 | 41.5 | 41.8 | 80.7 | 38.8 | 48.4 | 56.0 |
| Pass@k Training | 20.9/15.7 | 52.3 | 83.8 | 32.7 | 43.8 | 41.5 | 79.3 | 37.8 | 49.0 | 55.3 |
| **Our Methods** | | | | | | | | | | |
| DIVER-TD | 22.5/**16.9** | 59.4 | 82.2 | 27.9 | 44.7 | 42.3 | 83.4 | **42.3** | 49.5 | 58.4 |
| DIVER-ED | 20.9/15.7 | 59.7 | 84.0 | 31.6 | 46.1 | 43.0 | 83.4 | 36.2 | 49.9 | 56.5 |
| DIVER-MIX | 23.8/16.4 | **60.9** | **84.4** | 29.4 | 44.0 | **43.1** | **84.1** | 41.3 | 51.0 | **58.8** |

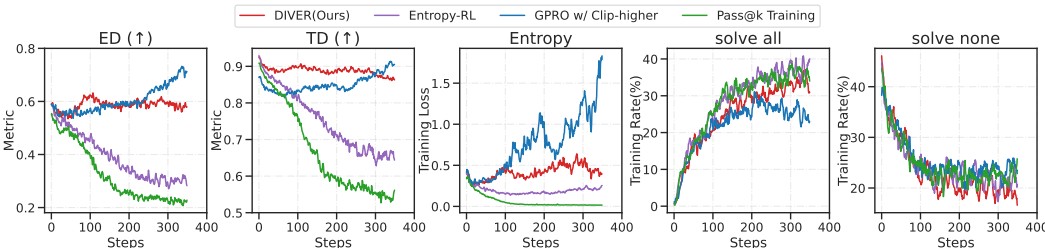

Figure 4: Training dynamics comparison with other exploration method across different metrics. ↑ indicates metrics where higher values are more diverse for ED and TD.

points on average. This advantage is especially evident on benchmarks that assess conversational capabilities, such as GPQA, where DIVER outperforms Entropy-RL by **+3.5** points. The relatively weaker performance of GPRO w/ Clip-higher indicates that merely increasing constraints without strategic direction fails to fundamentally enhance model generalization capabilities.

**Training Dynamics of Exploration Methods.** Figure. 4 illustrates the various metrics of DIVER compared to other exploration methods. We aim to maintain sufficient exploration (high ED and TD) while preventing excessive entropy growth that may lead to model collapse. With this logic in mind, we observe that Pass@k Training and Entropy-RL diversity metrics decline over time, indicating reduced exploration, while GPRO w/ Clip-higher maintains diversity but experiences problematic entropy increases later. In contrast, DIVER achieves optimal balance with high diversity and consistently reasonable entropy levels, enabling controlled exploration without excess randomness. Notably, DIVER's "solve all" rate grows more gradually, but its lower "solve none" rate in training demonstrates effective exploration without compromising solution quality.

**Extending DIVER to Different Models.** We further explore DIVER's adaptability across various language models, including *small, weak* or *different architecture* models. As shown in Figure.5 and Table 9, DIVER maintains effectiveness across different model backbones including Qwen2.5-Math-1.5B, Qwen2.5-7B-Base, and LLaMA-3.1-8B-Insturct. DIVER consistently outperforms baselines, improving over GRPO w/ Clip-higher by +1.7, +1.7, and +1.5 points on in-domain tasks respectively, with even larger gains of +1.5, +1.3, and +1.9 points on out-of-domain tasks. Ad-

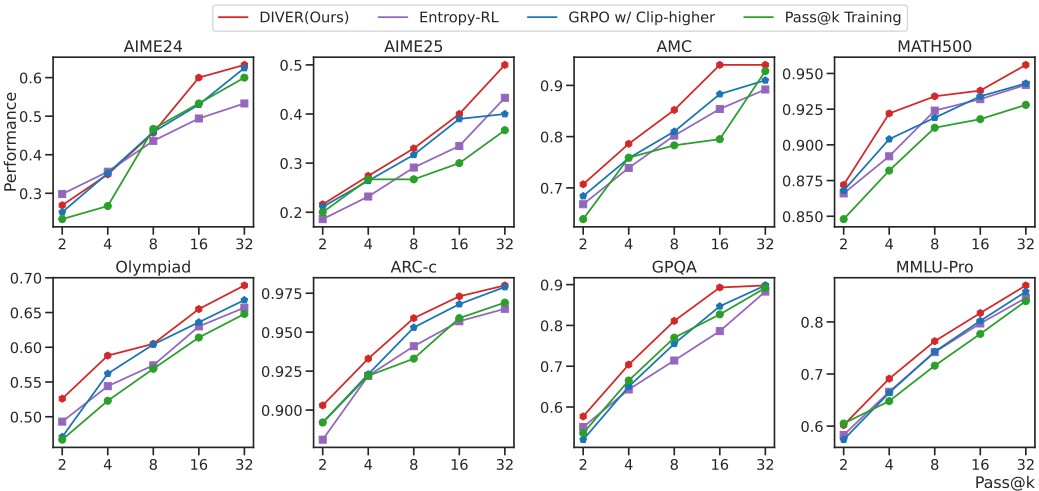

Figure 6: Comparison of different Pass@k performance across in-domain and out-of-domain benchmarks based on `Qwen2.5-Math-7B`. Complete results are available in Table 4.

ditionally, experiments with DeepSeek-R1-Distill-Qwen-7B, a model generating longer responses (2500-3500 tokens), show DIVER outperforming GPRO w/ Clip-higher by +1.6 points in-domain and +3.4 points out-of-domain, demonstrating its effectiveness with longer reasoning horizons.

### 4.3 Exploration Scope and Reasoning Capacity

We evaluate Pass@k, a metric for model reasoning upper-bounds (Cheng et al., 2025), for $k \in \{2, 4, 8, 16, 32\}$ across all tasks (Figure 6, Table 4). DIVER consistently outperforms baselines across nearly all benchmarks. Moreover, Pass@32 most closely reveals reasoning exploration scope, where DIVER achieves superior performance on challenging benchmarks with **50.0** on AIME25 (**+6.7** over Entropy-RL) and **68.9** points on OlympiadBench (**+2.1** over GRPO w/ Clip-higher). Overall, compared to existing exploration methods, DIVER uniquely enhances Pass@k capability without compromising Pass@1 performance. To visually demonstrate our conclusion, we examine case studies of multi-attempt rollouts (Appendix E). DIVER generates diverse yet coherent reasoning paths leading to correct solutions, while Entropy-RL explores at specific decision points, and GRPO w/ Clip-higher and Pass@k Training exhibit wide but unproductive exploration, all leading to incorrect answers. These results confirm DIVER's superior exploration scope unlocks higher reasoning capacity.

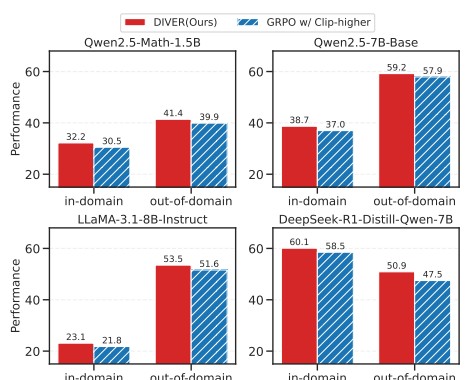

Figure 5: Average scores across in-domain and out-of-domain tasks with different models. Complete results in Table 9.

To address concerns about the scale of Pass@k, we extend our experiments to k=128, 256, 512, and 1024 across AIME24, AIME25, and AMC benchmarks, including the base model Qwen2.5-Math-7B and all baselines. The DIVER demonstrates consistent superiority of DIVER (table 5) across all settings: 1) DIVER achieves the highest Pass@k scores across nearly all benchmarks and k values, reaching 86.7% on AIME24 and 100% on AMC at Pass@1024. 2) The performance gap increases with larger k, highlighting DIVER's superior ability to generate diverse correct solutions. On AIME24, DIVER outperforms the second-best baseline by 6.7 points at Pass@1024. 3)All RL methods substantially improve over the base model, but DIVER consistently achieves the best results, confirming that diversity-driven exploration provides significant advantages.

## 4.4 ANALYSIS

**Mitigating Reward Hacking.** We compare four shaping strategies: applying diversity rewards to (1) correct responses, (2) all responses, (3) errors only, and (4) all responses with a length penalty. As shown in Figure 7, rewarding diversity on incorrect or all responses severely degrades test performance, with response lengths exploding as the model exploits long rollouts to gain higher diversity bonuses. Introducing a length penalty mitigates length explosion but still results in poor

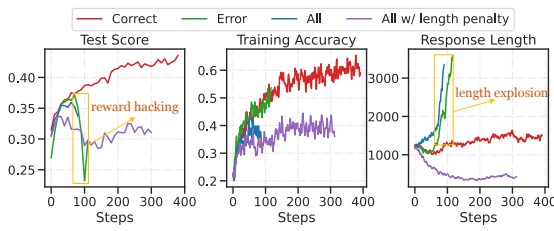

Figure 7: DIVER with rewarding diversity in correct, all, error, and all (w/ length penalty) responses.

accuracy. In contrast, the *conditional shaping* strategy, which applies diversity rewards only to correct responses (red line), effectively constrains exploration to valid solutions and mitigates reward hacking.

**Longer Horizons Improve Performance.** To investigate suitable exploration horizons (i.e., range of text for diversity calculation) for reasoning, we evaluate diversity metrics across different token horizons (i.e., the first 200, 500, 1000 tokens of the trajectory) versus complete responses. Figure 8 shows full responses maximize performance, while shorter horizons significantly reduce both entropy and performance. The consistent improvement with increasing horizon length confirms the reasoning benefits from global sequence-level diversity.

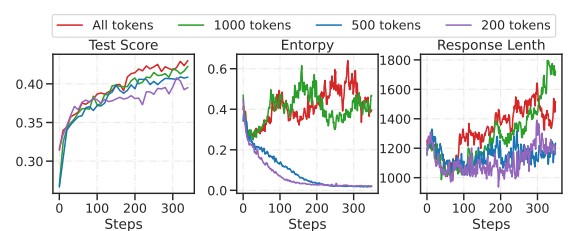

Figure 8: DIVER with varying horizon constraints. A longer horizon allows for higher global diversity.

**Diversity Enhances Reasoning Quality** To validate whether our approach achieves higher quality and more meaningful reasoning results, we evaluate all test responses using `deepseek-ai/DeepSeek-V3.2-Exp` as a judge model, ranking responses generated by the base model, baselines, and DIVER for each prompt. The evaluation comprehensively assesses four key dimensions: `Completeness`, `Logical Consistency`, `Relevance`, and `Efficiency`. Fig. 9 demonstrates DIVER solutions rank best (Rank 1) in 76.4% of cases, more than double the next best method. This confirms our diversity measures capture meaningful improvements in reasoning quality, not merely surface-level variations.

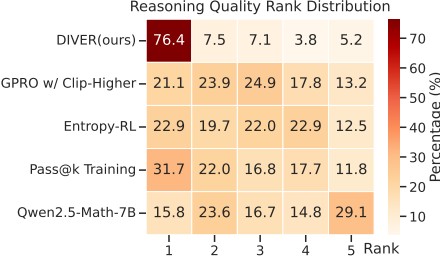

Figure 9: Average Ranking reasoning quality across six mathematical benchmarks as judged by DeepSeek-V3.2-Exp. Complete results are available in Table 8.

## 5 CONCLUSIONS, LIMITATIONS, AND FUTURE WORK

We presented DIVER, an innovative approach that enhances LLM reasoning from a new perspective of diversity. In contrast to existing methods that focus primarily on local token-level diversity, we examine the role of global sequence-level diversity in incentivizing deep exploration, revealing a positive correlation with reasoning capacity. Evaluations showed DIVER achieves consistently higher reasoning capabilities on in-domain tasks and stronger generalization on out-of-domain tasks. DIVER considers single-turn RLVR, while multi-turn settings hold greater promise for unlocking agent RL's potential in real-world applications. We leave it as future work. Another direction is to employ more powerful diversity metrics, such as using LLM-as-a-judge (Gu et al., 2024).

## ACKNOWLEDGEMENTS

We extend our gratitude to all the reviewers for their valuable feedback and suggestions. This work was completed during an internship at the Shanghai Artificial Intelligence Laboratory. It was supported in part by the National Natural Science Foundation of China (No. 62376122), and in part by the National Key Research and Development Program of China (No. 2025YFA1016904).

## ETHICS STATEMENT

We are not aware of any major ethical concerns arising from our work. Our study is conducted entirely within the mathematics domain, using only publicly available models and datasets for training and evaluation. No human subjects were involved, and our research does not introduce sensitive or potentially harmful insights.

## REPRODUCIBILITY STATEMENT

We provide the experimental setups in Sec.4.1, with with further details in Appendix B.2. The code is available at supplementary material, and will be released on GitHub in the future. Additionally, we will make the weights of the DIVER models publicly available via platforms such as Hugging Face Community upon acceptance.

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

# Appendix

# A  OPTIMAL POLICY INVARIANCE IN DIVER

Following the classical reward shaping study (Ng et al., 1999), we give the proof of Theorem 1, which guarantees the optimal policy invariance when incorporating global diversity as an intrinsic reward.

**Theorem 1** (Optimal Policy Invariance). *Let $M = (S, A, T, R, \gamma)$ denote the MDP for the LLM reasoning task. $d(\cdot) \colon S \mapsto \mathbb{R}$ is a real-valued function that computes the sentence-level diversity $d(s)$ of the state $s$ within a group of rollouts. We formulate $R_{int}(\cdot) \colon S \times A \times S \mapsto \mathbb{R}$ as an intrinsic reward function that is the difference between sentence diversities of two adjacent states, such that for all $s \in S, a \in A, s' \in S$, $R_{int}(s, a, s') = \gamma d(s') - d(s)$. Then, with any constant balancing ratio $\lambda$, every optimal policy in the transformed MDP $M' = (S, A, T, R + \lambda R_{int}, \gamma)$ will also be an optimal policy in $M$, and vice versa.*

*Proof.* For the original MDP $M$, we know that its optimal Q-function $Q_M^*$ satisfies the Bellman optimality equation (Sutton & Barto, 2018):

$$Q_M^*(s, a) = \mathbb{E}_{s'}\left[R(s, a, s') + \gamma \max_{a' \in A} Q_M^*(s', a')\right]. \tag{10}$$

With some simple algebraic manipulation, we can get:

$$Q_M^*(s, a) - \lambda d(s) = \mathbb{E}_{s'}\left[R(s, a, s') + \lambda\left(\gamma d(s') - d(s)\right) + \gamma \max_{a' \in A}\left(Q_M^*(s', a') - \lambda d(s')\right)\right]. \tag{11}$$

If we now define $\hat{Q}_{M'}(s, a) \triangleq Q_M^*(s, a) - \lambda d(s)$ and substitute that and $R_{int}(s, a, s') = \gamma d(s') - d(s)$ into the previous equation, we can get:

$$\begin{aligned}
\hat{Q}_{M'}(s, a) &= \mathbb{E}_{s'}\left[R(s, a, s') + \lambda R_{int}(s, a, s') + \gamma \max_{a' \in A} \hat{Q}_{M'}(s', a')\right] \\
&= \mathbb{E}_{s'}\left[R'(s, a, s') + \gamma \max_{a' \in A} \hat{Q}_{M'}(s', a')\right],
\end{aligned} \tag{12}$$

which is exactly the Bellman optimality equation for the transformed MDP $M'$, where $R' = R + \lambda R_{int}$ is the reward function for $M'$. Thus, $Q_{M'}^*(s, a) = \hat{Q}_{M'}(s, a) = Q_M^*(s, a) - \lambda d(s)$, and the optimal policy for $M'$ therefore satisfies:

$$\begin{aligned}
\pi_{M'}^*(s) &= \arg \max_{a \in A} Q_{M'}^*(s, a) \\
&= \arg \max_{a \in A}\left[Q_M^*(s, a) - \lambda d(s)\right] \\
&= \arg \max_{a \in A} Q_M^*(s, a),
\end{aligned} \tag{13}$$

and is therefore also optimal in $M$. To show every optimal policy in $M$ is also optimal in $M'$, simply apply the same proof with the roles of $M$ and $M'$ interchanged (and using $-R_{int}$ as the intrinsic reward). This completes the proof. $\square$

## B EXPERIMENTAL DETAILS

### B.1 DIVERSITY BASED FILTERING

To compare the model training process using rollouts with different diversity levels, we conduct an illustrative experiment to filter GRPO rollouts into high-diversity and low-diversity subsets to train separate models. Specifically, the policy generates $2*G$ responses for each query. For high-diversity scheme, $G$ responses of highest diversity metrics (TD and ED) are filtered as the candidate responses $o_1, ..., o_G$ for GRPO training. For low-diversity scheme, responses of lowest diversity metrics are filtered. Both approaches maintain identical computational costs, as they use the same number of generated responses and differ only in the selection criteria applied during filtering.

### B.2 DETAILED SETTINGS

**Datasets**    Our training data is a subset of OpenR1-Math-220k (Face, 2025), with prompts collected from NuminaMath 1.5 (LI et al., 2024). We follow the LUFFY (**?**)[1] dataset construction methodology but differ in that we do not incorporate off-policy reasoning traces, as ours is a purely on-policy approach.

**Evaluation**    We evaluate our method on six mathematical reasoning benchmarks: AIME 2024[2], AIME 2025[3], AMC (Li et al., 2024), Minerva (Lewkowycz et al., 2022), OlympiadBench (He et al., 2024), and MATH-500 (Hendrycks et al., 2021). Our main results report avg@32 for the smaller test sets (AIME 2024, AIME 2025, AMC) and pass@1 for the larger benchmarks. For Pass@k evaluation, we generate $k$ completions and select the one with the highest reward score. For cross-domain generalization, we test on ARC-c (Clark et al., 2018), GPQA-diamond (GPQA$^*$) (Rein et al., 2024), and MMLU-Pro.

**RL Practice**    We set $\beta = 0$ to remove the KL loss term and use 0.28 for higher clip following GPPO w/ Clip-higher. Detailed implementation parameters are provided in Table 3. All training experiments are conducted using 8 A100 GPUs. We train 350 steps Qwen2.5-Math-7B, and 200 steps for others. Our implementation is based on verl[4], which uses vLLM[5] as the rollout generators. We are thankful for these open-source repositories.

Table 2: Computation overhead analysis.

|  | DIVER(ours) | GRPO w/ Clip-Higher | Entropy-RL | Pass@k Training |
|---|---|---|---|---|
| Time Cost (350 Steps) | 29.75h | 28.46h | 28.13h | 30.73h |

The computation overhead of calculating textual diversity and equational diversity is negligible, since both metrics only involve rule-based calculations (e.g., n-gram matching for BLEU score or string recognition for equation extraction) without any feedforward or backpropagation of large-scale models. As shown in Table 2, DIVER only incurs a 5% increase in training time compared to GRPO and Entropy-RL baselines, while requiring less training time than Pass@k Training baseline.

### B.3 SYSTEM PROMPT

We use the same system prompt for training and inference in all our models except LLaMA-3.1-8B-Instruct:

---

[1] `https://huggingface.co/datasets/Elliott/Openr1-Math-46k-8192`
[2] `https://huggingface.co/datasets/HuggingFaceH4/aime_2024`
[3] `https://huggingface.co/datasets/PrimeIntellect/AIME-25`
[4] https://github.com/volcengine/verl
[5] https://github.com/vllm-project/vllm

Table 3: Hyperparameter settings

| Hyperparameter | Value | Hyperparameter | Value |
|---|---|---|---|
| max prompt length | 1024 | KL coefficient $\beta$ | 0.0 |
| max response length | 8192 | train temperature | 1.0 |
| num generations $G$ | 8 | eval temperature | 0.6 |
| gpu memory utilization | 0.85 | entropy coefficient | 0.0 |
| learning rate | 1e-6 | high clip ratio $\epsilon_h$ | 0.28 |
| train batch size | 128 | low clip ratio $\epsilon_l$ | 0.20 |
| mini batch size | 32 | shaping ratio $\lambda$ | 0.1 |
| use dynamic batch size | True | diversity upper bound $\sigma$ | 0.65 |
| validate batch size | 512 | | |

---

Your task is to follow a systematic, thorough reasoning process before providing the final solution. This involves analyzing, summarizing, exploring, reassessing, and refining your thought process through multiple iterations. Structure your response into two sections: Thought and Solution. In the Thought section, present your reasoning using the format: "`<think>`\n thoughts `</think>`\n". Each thought should include detailed analysis, brainstorming, verification, and refinement of ideas. After "`</think>`\n" in the Solution section, provide the final, logical, and accurate answer, clearly derived from the exploration in the Thought section. If applicable, include the answer in `\boxed{}` for closed-form results like multiple choices or mathematical solutions.
**User:** This is the problem: {QUESTION}
**Assistant:** `<think>`

---

For LLaMA-3.1-8B-Instruct, we use a simplified prompt which only includes the CoT prompt:

---

**User:** {QUESTION}
**Answer:** Let's think step by step.

---

## C    DETAILED METRICS DEFINITION

### C.1    PASS@k PERFORMANCE

Given a question $x$, we employ the model to generate $k$ independent and identically distributed (i.i.d.) responses. Each response is evaluated by a binary reward function, yielding $r_i \in 0, 1$ where $r_i = 1$ indicates a correct response. The pass@$k$ metric quantifies the probability of obtaining at least one correct response among the $k$ samples:

$$\text{pass@k} = \mathbb{P}\left[\bigvee_{i=1}^{k}(r_i = 1)\right] = \mathbb{E}\left[1 - \prod_{i=1}^{k}(1 - r_i)\right] \tag{14}$$

While pass@1 evaluates evaluates the model's accuracy on first attempts, pass@$k$ metric emphasizes the model's ability to generate diverse solutions and improve success rates through sampling. To rigorously demonstrate DIVER's consistent advantage, we repeat all experiments across Pass@2 to Pass@32, and report the means and standard deviations for a total of 3 runs. As shown in figure 6 and table 4, our method consistently outperforms all baselines approaches (GRPO w/ Clip-higher, Entropy-RL, and Pass@k Training) across the spectrum of pass@$k$ metrics (k=2 to k=32) on both in-distribution benchmarks and out-of-distribution benchmarks. The performance gap is particularly significant at higher k values, where DIVER demonstrates superior exploration capabilities and achieves the highest average scores.

Table 4: pass@k performance.

| Pass@k | Method | In-Distribution Performance | | | | | | Out-of-Distribution Performance | | | |
|---|---|---|---|---|---|---|---|---|---|---|---|
| | | AIME 24/25 | AMC | MATH-500 | Minerva | Olympiad | Avg. | ARC-c | GPQA* | MMLU-Pro | Avg. |
| $k=2$ | Qwen2.5-Math-7B | $23.3 \pm 3.3/12.2 \pm 5.1$ | $55.4 \pm 1.2$ | $74.7 \pm 0.8$ | $27.2 \pm 1.3$ | $35.5 \pm 0.8$ | $38.1 \pm 0.3$ | 89.2 | 52.0 | 57.4 | 66.2 |
| | GPRO w/ Clip-higher | $26.6 \pm 5.8/18.9 \pm 1.9$ | $61.4 \pm 3.7$ | $85.7 \pm 1.5$ | $38.0 \pm 0.2$ | $50.4 \pm 0.7$ | $46.8 \pm 1.3$ | 89.2 | 52.0 | 57.4 | 66.2 |
| | Entropy-RL | $26.7 \pm 0.0/22.2 \pm 3.9$ | $63.9 \pm 1.2$ | $86.3 \pm 0.5$ | $35.5 \pm 0.8$ | $50.5 \pm 2.4$ | $47.5 \pm 0.7$ | 88.1 | 55.1 | 58.3 | 67.2 |
| | Pass@k Training | $23.3 \pm 3.4/22.2 \pm 6.9$ | $63.5 \pm 0.7$ | $85.1 \pm 1.0$ | $\mathbf{40.7 \pm 1.2}$ | $47.6 \pm 1.6$ | $47.1 \pm 1.7$ | 89.2 | 53.6 | $\mathbf{60.5}$ | 67.8 |
| | **DIVER(ours)** | $\mathbf{28.9 \pm 5.1/27.8 \pm 8.4}$ | $\mathbf{67.9 \pm 1.4}$ | $\mathbf{87.7 \pm 0.1}$ | $42.9 \pm 1.1$ | $\mathbf{52.1 \pm 1.5}$ | $\mathbf{51.2 \pm 1.9}$ | $\mathbf{90.3}$ | $\mathbf{57.7}$ | 60.2 | $\mathbf{69.4}$ |
| $k=4$ | Qwen2.5-Math-7B | $30.0 \pm 5.8/16.6 \pm 5.8$ | $67.1 \pm 3.5$ | $81.5 \pm 0.6$ | $35.5 \pm 0.6$ | $43.8 \pm 0.5$ | $45.8 \pm 1.7$ | 92.3 | 65.0 | 66.4 | 74.6 |
| | GPRO w/ Clip-higher | $32.2 \pm 11.7/25.5 \pm 3.9$ | $72.3 \pm 1.2$ | $88.9 \pm 1.1$ | $\mathbf{47.3 \pm 1.1}$ | $53.3 \pm 0.9$ | $53.2 \pm 0.9$ | 92.3 | 65.0 | 66.4 | 74.6 |
| | Entropy-RL | $\mathbf{38.9 \pm 3.8/28.9 \pm 3.8}$ | $74.3 \pm 2.8$ | $89.5 \pm 0.1$ | $40.6 \pm 1.2$ | $54.7 \pm 0.8$ | $54.5 \pm 1.3$ | 92.2 | 64.3 | 66.6 | 74.4 |
| | Pass@k Training | $36.7 \pm 3.4/24.4 \pm 8.4$ | $71.5 \pm 3.0$ | $88.5 \pm 0.2$ | $42.2 \pm 1.1$ | $55.7 \pm 0.7$ | $53.2 \pm 1.6$ | 92.2 | 66.5 | 64.8 | 74.5 |
| | **DIVER(ours)** | $37.8 \pm 1.9/24.4 \pm 3.9$ | $\mathbf{75.9 \pm 4.3}$ | $\mathbf{90.5 \pm 0.5}$ | $47.3 \pm 1.9$ | $\mathbf{57.7 \pm 0.4}$ | $\mathbf{55.6 \pm 1.6}$ | $\mathbf{93.3}$ | $\mathbf{70.4}$ | $\mathbf{69.1}$ | $\mathbf{77.6}$ |
| $k=8$ | Qwen2.5-Math-7B | $42.2 \pm 5.1/23.3 \pm 3.4$ | $77.1 \pm 2.1$ | $86.7 \pm 1.2$ | $41.3 \pm 0.6$ | $51.3 \pm 0.9$ | $53.7 \pm 1.1$ | 95.3 | 75.5 | 74.3 | 81.7 |
| | GPRO w/ Clip-higher | $35.6 \pm 5.1/30.0 \pm 5.8$ | $77.9 \pm 1.8$ | $90.7 \pm 0.3$ | $\mathbf{52.1 \pm 1.3}$ | $58.5 \pm 2.1$ | $57.4 \pm 1.1$ | 95.3 | 75.5 | 74.3 | 81.7 |
| | Entropy-RL | $41.1 \pm 1.9/28.9 \pm 3.8$ | $80.3 \pm 2.5$ | $91.9 \pm 0.8$ | $45.6 \pm 1.0$ | $59.9 \pm 0.5$ | $58.0 \pm 1.2$ | 94.1 | 71.4 | 74.2 | 79.9 |
| | Pass@k Training | $36.7 \pm 3.4/24.4 \pm 2.0$ | $80.3 \pm 0.7$ | $90.7 \pm 0.1$ | $45.5 \pm 1.9$ | $60.2 \pm 0.4$ | $56.3 \pm 0.6$ | 93.3 | 77.0 | 71.6 | 80.6 |
| | **DIVER(ours)** | $\mathbf{45.6 \pm 1.9/30.0 \pm 0.0}$ | $\mathbf{83.9 \pm 2.5}$ | $\mathbf{92.9 \pm 0.1}$ | $50.6 \pm 1.1$ | $\mathbf{61.8 \pm 0.7}$ | $\mathbf{60.8 \pm 0.0}$ | $\mathbf{95.9}$ | $\mathbf{81.1}$ | $\mathbf{76.3}$ | $\mathbf{84.4}$ |
| $k=16$ | Qwen2.5-Math-7B | $43.3 \pm 12.0/23.3 \pm 3.4$ | $81.1 \pm 3.0$ | $89.9 \pm 0.3$ | $49.1 \pm 0.8$ | $56.5 \pm 1.5$ | $56.0 \pm 2.2$ | 96.8 | 84.7 | 80.2 | 87.2 |
| | GPRO w/ Clip-higher | $45.6 \pm 5.1/37.8 \pm 5.1$ | $84.3 \pm 2.4$ | $92.3 \pm 0.3$ | $\mathbf{59.3 \pm 2.2}$ | $63.2 \pm 0.5$ | $63.7 \pm 2.3$ | 96.8 | 84.7 | 80.2 | 87.2 |
| | Entropy-RL | $47.8 \pm 5.1/35.6 \pm 2.0$ | $\mathbf{88.0 \pm 2.1}$ | $93.7 \pm 0.2$ | $50.7 \pm 1.6$ | $62.8 \pm 0.3$ | $63.2 \pm 0.6$ | 95.7 | 78.6 | 79.7 | 84.7 |
| | Pass@k Training | $\mathbf{50.0 \pm 6.7}/34.4 \pm 8.4$ | $82.7 \pm 2.5$ | $92.3 \pm 0.1$ | $50.6 \pm 1.8$ | $64.0 \pm 0.5$ | $62.3 \pm 0.2$ | 95.9 | 82.7 | 77.7 | 85.4 |
| | **DIVER(ours)** | $47.8 \pm 5.1/\mathbf{38.9 \pm 3.8}$ | $\mathbf{88.0 \pm 0.0}$ | $\mathbf{94.4 \pm 0.4}$ | $56.5 \pm 2.9$ | $\mathbf{66.0 \pm 0.7}$ | $\mathbf{65.3 \pm 1.0}$ | $\mathbf{97.3}$ | $\mathbf{89.3}$ | $\mathbf{81.7}$ | $\mathbf{89.4}$ |
| $k=32$ | Qwen2.5-Math-7B | $52.2 \pm 5.1/30.0 \pm 3.3$ | $88.8 \pm 3.5$ | $92.1 \pm 0.5$ | $54.4 \pm 1.0$ | $62.2 \pm 0.8$ | $63.3 \pm 0.9$ | 97.9 | 89.8 | 85.8 | 91.2 |
| | GPRO w/ Clip-higher | $56.7 \pm 10.0/42.2 \pm 3.9$ | $92.0 \pm 2.8$ | $94.4 \pm 0.4$ | $54.9 \pm 0.4$ | $65.8 \pm 0.6$ | $67.6 \pm 0.8$ | 97.9 | 89.8 | 85.8 | 91.2 |
| | Entropy-RL | $54.4 \pm 2.0/42.2 \pm 1.9$ | $90.9 \pm 3.0$ | $94.1 \pm 0.3$ | $51.4 \pm 0.3$ | $66.3 \pm 0.5$ | $67.7 \pm 0.6$ | 96.5 | 88.3 | 84.6 | 89.8 |
| | Pass@k Training | $56.7 \pm 5.8/33.3 \pm 6.7$ | $88.4 \pm 0.7$ | $93.8 \pm 0.3$ | $53.8 \pm 1.5$ | $68.0 \pm 0.5$ | $65.7 \pm 1.7$ | 96.9 | 89.2 | 84.0 | 90.0 |
| | **DIVER(ours)** | $\mathbf{58.9 \pm 3.8/48.9 \pm 1.9}$ | $\mathbf{92.4 \pm 1.9}$ | $\mathbf{95.4 \pm 0.2}$ | $64.4 \pm 3.9$ | $\mathbf{69.0 \pm 0.3}$ | $\mathbf{70.5 \pm 0.1}$ | $\mathbf{98.0}$ | $\mathbf{89.8}$ | $\mathbf{87.0}$ | $\mathbf{91.6}$ |

### C.2    BLEU SCORE

BLUE measures the similarity between a candidate string $c$ and a reference string $r$ by calculating the $n$-gram (short phrases of length $n$) precision, while also penalizing overly short outputs through

Table 5: Performance comparison with larger pass@k values.

| Benchmark | Method | pass@128 | pass@256 | pass@512 | pass@1024 |
|---|---|---|---|---|---|
| AIME24 | Qwen2.5-Math-7B | 63.3 | 70.0 | 76.7 | 80.0 |
| | GPRO w/ Clip-Higher | 70.0 | 73.3 | 76.7 | 76.7 |
| | Entropy-RL | 70.0 | 73.3 | 76.7 | 76.7 |
| | Pass@k Training | 70.0 | 76.7 | 80.0 | 80.0 |
| | **DVIER(ours)** | **76.7** | **80.0** | **83.3** | **86.7** |
| AIME25 | Qwen2.5-Math-7B | 40.0 | 56.7 | 63.3 | 66.7 |
| | GPRO w/ Clip-Higher | 60.0 | 63.3 | 66.7 | 70.0 |
| | Entropy-RL | **63.3** | 63.3 | 66.7 | 73.3 |
| | Pass@k Training | **63.3** | 63.3 | **70.0** | 73.3 |
| | **DVIER(ours)** | **63.3** | **66.7** | **70.0** | **76.7** |
| AMC | Qwen2.5-Math-7B | 92.8 | **97.6** | 97.6 | 97.6 |
| | GPRO w/ Clip-Higher | 94.0 | 96.4 | 97.6 | 98.8 |
| | Entropy-RL | **96.4** | 96.4 | 97.6 | 98.8 |
| | Pass@k Training | 95.2 | 96.4 | 97.6 | **100.0** |
| | **DVIER(ours)** | **96.4** | **97.6** | **98.8** | **100.0** |

a brevity penalty as

$$\text{BLEU} = \text{BP} \cdot \exp\left(\sum_{n=1}^{N} w_n \log p_n\right), \quad p_n = \frac{\sum_{g \in G_n} \min\{C_c(g), C_r(g)\}}{\sum_{g \in G_n} C_c(g)}, \tag{15}$$

where $N$ is the maximum $n$-gram length considered, $p_n$ is the modified precision for $n$-grams of size $n$, $w_n$ is the weight for each $n$-gram level (usually uniform, e.g., $w_n = 1/N$), $G_n$ is the set of $n$-grams, and $C_c(g)/C_r(g)$ counts how often the $n$-gram $g$ appears in string $c/r$. The brevity penalty BP is defined as

$$\text{BP} = \begin{cases} 1, & |c| \geq |r|, \\ \exp(1 - |r|/|c|), & \text{otherwise.} \end{cases} \tag{16}$$

## C.3 EQUATION EXTRACTION

We extract mathematical formulas using regular expressions that identify three common LaTeX notation patterns: `\\[ \\]`, `\\( \\)`, and `$ $`. Our implementation employs `re.findall()` to capture these patterns and stores them in a set structure to eliminate duplicates.

To evaluate the extraction accuracy, we compare the automatically extracted equation counts against human-annotated ground-truth values. For demonstration, we randomly sample 5 responses from different query inputs. Table 6 presents the equation counts obtained through our extraction method alongside the corresponding ground-truth values. The observed failure rate is approximately 8.4%(8/95).

We note that since Equational Diversity (ED) serves as an intrinsic reward signal to encourage exploration during training, perfect extraction accuracy is not required. The metric only needs to approximate response diversity with sufficient fidelity to guide effective exploration. The observed accuracy level is adequate for this purpose, as evidenced by the consistent performance improvements shown in our main results.

Table 6: Equation extraction accuracy evaluation on randomly sampled responses.

| | Response 1 | Response 2 | Response 3 | Response 4 | Response 5 |
|---|---|---|---|---|---|
| No. of equations via extraction | 14 | 25 | 13 | 17 | 18 |
| Ground-truth value | 17 | 25 | 18 | 17 | 18 |

# D MORE ANALYZE

## D.1 ANALYSIS OF DIVERSITY METRICS VALIDITY

To verify whether TD and ED truly capture semantic-level reasoning diversity, we conduct a correlation analysis comparing them with two semantic similarity metrics: 1) LLM-based scores: We compute the cosine similarity of the model's hidden states from the final layer outputs using `Qwen2.5-7B-Instruct` for all baselines and our method; 2) Embedding-based scores: We use `google/embeddinggemma-300m` to encode responses into embeddings and compute their cosine similarity.

For each method (DIVER and baselines), we generate multiple responses per prompt and compute all diversity metrics within each response group. Evaluated on 1,560 rollouts total, the average metrics are shown in the table 7 (↑ indicates higher values represent greater diversity; ↓ indicates the opposite). The results demonstrate strong consistency between TD/ED and LLM-/Embedding-based alternatives. Methods achieving higher TD/ED scores consistently exhibit lower LLM-/Embedding-based semantic similarity scores, confirming that our choice of diversity metrics effectively captures meaningful semantic diversity rather than merely surface-level variation. This validates the generality of TD and ED as reliable proxies for reasoning diversity.

Table 7: Correlation between diversity metrics across different methods. Higher TD/ED (↑) and lower LLM/Embedding similarity scores (↓) indicate greater diversity.

| Method | TD (↑) | ED (↑) | LLM-based (↓) | Embedding-based (↓) | pass@1 | pass@8 |
|---|---|---|---|---|---|---|
| **DIVER(ours)** | 0.702 | 0.477 | 0.936 | 0.881 | 43.1 | 60.8 |
| GRPO w/ Clip-Higher | 0.805 | 0.575 | 0.916 | 0.841 | 40.7 | 57.4 |
| Entropy-RL | 0.549 | 0.294 | 0.947 | 0.954 | 41.8 | 58.0 |
| Pass@k Training | 0.441 | 0.200 | 0.959 | 0.967 | 41.5 | 56.3 |

Moreover, our diversity metrics are both computationally lightweight and easy to implement. The computation overhead of calculating textual diversity and equational diversity is negligible, since both metrics only involve rule-based calculations (e.g., n-gram matching for BLEU score or string recognition for equation extraction) without any feedforward or backpropagation of large-scale models (e.g., the above LLM-/Embedding-based metrics).

## D.2 ANALYSIS OF REASONING QUALITY

To validate whether our approach achieves higher quality and more meaningful reasoning results, we evaluate all test responses using `deepseek-ai/DeepSeek-V3.2-Exp` as a judge model, ranking responses generated by the base model, baselines, and DIVER for each prompt. The evaluation comprehensively assesses four key dimensions: `Completeness`, `Logical Consistency`, `Relevance`, and `Efficiency`. The table 8 presents the final result over all six benchmarks: AIME24/25, AMC, MATH-500, Minerva, and Olympiad. It demonstrates DIVER's consistent superiority, achieving a 1.52 average ranking compared to baseline methods ranging from 2.57 to 3.19. DIVER solutions rank best (Rank 1) in 76.4% of cases—more than double the next best method. This confirms our diversity measures capture meaningful improvements in reasoning quality, not merely surface-level variations.

## D.3 ANALYSIS OF MODEL SCALE AND REASONING HORIZON

We further explore DIVER's adaptability across various language models, including *small, weak* or *different architecture* models. As shown in Figure.5, DIVER consistently outperforms baselines across all model variants, demonstrating its strong generalization capabilities.

To further validate DIVER's effectiveness on models with longer reasoning horizons, we conducted additional experiments using DeepSeek-R1-Distill-Qwen-7B as the base model, which typically generates much longer responses (2500-3500 tokens). The results in Table 9 show that DIVER maintains its consistent superiority even with these extended reasoning processes, achieving 60.1% average in-domain performance and 50.9% out-of-domain performance, compared to GPRO w/

Table 8: Ranking reasoning quality across six mathematical benchmarks as judged by DeepSeek-V3.2-Exp. Overall Avg. shows aggregate performance and lower Avg. Rank($\downarrow$) is better.

| Benchmark | Model | Avg. Rank($\downarrow$) | Rank 1(best) | Rank2 | Rank 3 | Rank4 | Rank5 (worst) |
|---|---|---|---|---|---|---|---|
| Overall Avg. | Qwen2.5-Math-7B | 3.19 | 15.8% | 23.6% | 16.7% | 14.8% | 29.1% |
| | GPRO w/ Clip-Higher | 2.77 | 21.1% | 23.9% | 24.9% | 17.8% | 13.2% |
| | Entropy-RL | 2.86 | 22.9% | 19.7% | 22.0% | 22.9% | 12.5% |
| | Pass@k Training | 2.57 | 31.7% | 22.0% | 16.8% | 17.7% | 11.8% |
| | **DIVER(ours)** | **1.52** | **76.4%** | 7.5% | 7.1% | 3.8% | 5.2% |
| AIME24 | Qwen2.5-Math-7B | 3.53 | 6.7% | 23.3% | 16.7% | 16.7% | 36.7% |
| | GPRO w/ Clip-Higher | 3.13 | 6.7% | 30.0% | 23.3% | 23.3% | 16.7% |
| | Entropy-RL | 3.47 | 6.7% | 13.3% | 30.0% | 26.7% | 23.3% |
| | Pass@k Training | 2.73 | 26.7% | 16.7% | 20.0% | 30.0% | 6.7% |
| | **DIVER(ours)** | **1.77** | **66.7%** | 6.7% | 16.7% | 3.3% | 6.7% |
| AIME25 | Qwen2.5-Math-7B | 3.73 | 0.0% | 30.0% | 10.0% | 16.7% | 43.3% |
| | GPRO w/ Clip-Higher | 2.83 | 16.7% | 26.7% | 23.3% | 23.3% | 10.0% |
| | Entropy-RL | 3.00 | 20.0% | 13.3% | 23.3% | 33.3% | 10.0% |
| | Pass@k Training | 2.70 | 26.7% | 20.0% | 26.7% | 10.0% | 16.7% |
| | **DIVER(ours)** | **1.83** | **60.0%** | 13.3% | 16.7% | 3.3% | 6.7% |
| AMC | Qwen2.5-Math-7B | 3.17 | 20.5% | 18.1% | 15.7% | 15.7% | 30.1% |
| | GPRO w/ Clip-Higher | 2.73 | 21.7% | 20.5% | 31.3% | 15.7% | 10.8% |
| | Entropy-RL | 2.82 | 26.5% | 18.1% | 18.1% | 21.7% | 15.7% |
| | Pass@k Training | 2.49 | 31.3% | 25.3% | 16.9% | 15.7% | 10.8% |
| | **DIVER(ours)** | **1.27** | **85.5%** | 8.4% | 1.2% | 3.6% | 1.2% |
| MATH-500 | Qwen2.5-Math-7B | 2.41 | 39.8% | 21.6% | 13.0% | 9.0% | 16.6% |
| | GPRO w/ Clip-Higher | 2.27 | 43.8% | 18.2% | 16.2% | 11.0% | 10.8% |
| | Entropy-RL | 2.24 | 43.4% | 20.0% | 13.6% | 14.8% | 8.2% |
| | Pass@k Training | 2.19 | 48.2% | 16.2% | 13.4% | 12.6% | 9.6% |
| | **DIVER(ours)** | **1.40** | **82.6%** | 5.0% | 4.8% | 4.8% | 2.8% |
| Minerva | Qwen2.5-Math-7B | 3.14 | 12.1% | 26.8% | 21.3% | 14.3% | 25.4% |
| | GPRO w/ Clip-Higher | 2.93 | 17.6% | 22.8% | 24.3% | 19.5% | 15.8% |
| | Entropy-RL | 2.79 | 18.0% | 25.4% | 26.1% | 20.6% | 9.9% |
| | Pass@k Training | 2.70 | 25.4% | 25.0% | 16.2% | 21.3% | 12.1% |
| | **DIVER(ours)** | **1.41** | **82.4%** | 5.9% | 4.4% | 2.6% | 4.8% |
| Olympiad | Qwen2.5-Math-7B | 3.18 | 15.7% | 22.1% | 18.5% | 16.2% | 27.6% |
| | GPRO w/ Clip-Higher | 2.72 | 20.4% | 25.4% | 28.0% | 14.0% | 12.1% |
| | Entropy-RL | 2.81 | 21.1% | 22.8% | 20.9% | 24.2% | 10.9% |
| | Pass@k Training | 2.60 | 25.9% | 28.5% | 17.3% | 16.4% | 11.9% |
| | **DIVER(ours)** | **1.45** | **81.2%** | 5.9% | 4.0% | 4.8% | 4.0% |

Table 9: Overall performance on six competition-level benchmark performance on Qwen2.5-Math-1.5B, Qwen2.5-7B-Base, LLaMA-3.1-8B-Instruct and DeepSeek-R1-Distill-Qwen-7B.

| Model | In-Distribution Performance | | | | | | Out-of-Distribution Performance | | | |
|---|---|---|---|---|---|---|---|---|---|---|
| | AIME 24/25 | AMC | MATH-500 | Minerva | Olympiad | Avg. | ARC-c | GPQA* | MMLU-Pro | Avg. |
| *Qwen2.5-Math-1.5B* | | | | | | | | | | |
| GRPO w/ Clip-higher | 9.9/7.7 | 42.8 | 68.6 | 20.6 | 33.4 | 30.5 | 59.5 | 30.1 | 30.2 | 39.9 |
| **DIVER** | 11.0/8.3 | 44.3 | 71.8 | 23.9 | 33.6 | 32.2 | 33.2 | 59.1 | 31.8 | 41.4 |
| *Qwen2.5-7B-Base* | | | | | | | | | | |
| GRPO w/ Clip-higher | 15.8/12.1 | 49.2 | 78.2 | 27.6 | 39.1 | 37.0 | 89.8 | 28.6 | 55.4 | 57.9 |
| **DIVER** | 20.9/12.9 | 48.9 | 76.0 | 30.9 | 42.8 | 38.7 | 91.1 | 31.1 | 55.2 | 59.2 |
| *LLaMA-3.1-8B-Instruct* | | | | | | | | | | |
| GRPO w/ Clip-higher | 6.9/1.4 | 23.3 | 53.0 | 23.5 | 22.0 | 21.8 | 89.2 | 14.8 | 50.8 | 51.6 |
| **DIVER** | 7.7/1.5 | 28.8 | 56.4 | 22.8 | 21.7 | 23.1 | 88.2 | 20.4 | 52.0 | 53.5 |
| *DeepSeek-R1-Distill-Qwen-7B* | | | | | | | | | | |
| GPRO w/ Clip-Higher | 49.3/35.0 | 81.7 | 90.2 | 35.7 | 59.0 | 58.5 | 85.2 | 2.6 | 54.9 | 47.5 |
| **DIVER(ours)** | 51.1/36.9 | 82.0 | 92.6 | 36.8 | 61.2 | 60.1 | 85.6 | 10.7 | 56.6 | 50.9 |

Clip-Higher's 58.5% and 47.5% respectively. This confirms that our approach scales effectively to longer-horizon reasoning scenarios.

## D.4 MORE COMPREHENSIVE ABLATIONS

**Analysis of diversity reward shaping coefficient.** We further analyze the sensitivity of DIVER to the diversity reward shaping coefficient $\lambda$. Fig. 10 demonstrates performance remains stable across different $\lambda$ values, with test score curves following similar trajectories regardless of coefficient magnitude. Even when $\lambda$ becomes relatively large, we observe only minor performance degradation without triggering reward hacking. This robustness to hyperparameter selection makes DIVER practical for real-world applications, as it doesn't require precise tuning of diversity reward weights.

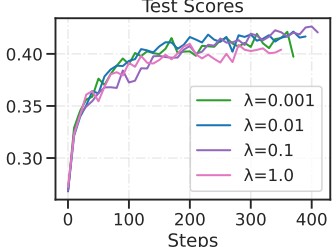

Figure 10: Average test scores with varying coefficients ($\lambda$).

**Longer Horizons Improve Performance.** It is important to clarify that the "horizon" in our analysis refers to the window size used for calculating diversity metrics, rather than the full response length, in Fig.11. For instance, when we use a horizon of 200, we compute Textual Diversity (TD) and Equational Diversity (ED) metrics only on the first 200 tokens of a generated response, even though the complete response may be much longer (1600-1800 tokens in our main experiments).

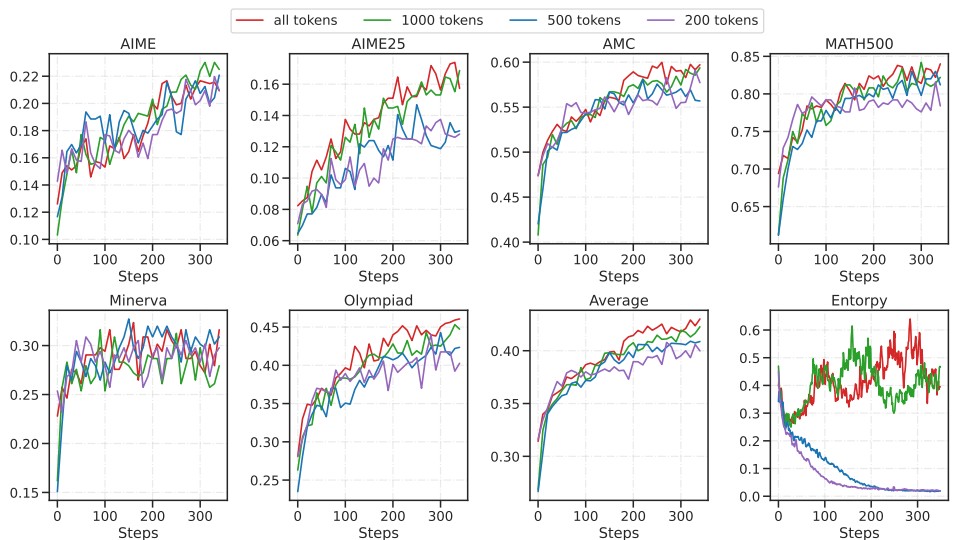

Figure 11: DIVER performance under different token horizon constraints.

**Hyperparameter Robustness.** To analyze hyperparameter sensitivity, we evaluate DIVER and baseline across different group sizes and training temperatures. As shown in Table 10, DIVER demonstrates superior robustness to hyperparameter variations. When reducing group size from G = 8 to G = 4, DIVER shows a smaller relative performance decline than GPRO w/ Clip-Higher, while maintaining a 2.5 point advantage. Similarly, when decreasing temperature from t = 1.0 to t = 0.6, DIVER exhibits minimal performance fluctuation (0.3 point drop) compared to the baseline's more significant 1.2 point reduction. This stability across different configuration settings highlights DIVER's reliability in practical scenarios where optimal hyperparameter tuning may be constrained.

Table 10: Performance comparison across different hyperparameters.

| Hyperparameter | Model | AIME 24 | AIME 25 | AMC | MATH-500 | Minerva | Olympiad | Avg. |
|---|---|---|---|---|---|---|---|---|
| | | *Group Size G* | | | | | | |
| G = 8 | GPRO w/ Clip-Higher | 18.9 | 16.4 | 57.3 | 81.2 | 28.7 | 41.5 | 40.7 |
| | **DIVER** | 23.8 | 16.4 | 60.9 | 84.4 | 29.4 | 44.0 | 43.1 |
| G = 4 | GPRO w/ Clip-Higher | 16.3 | 12.1 | 55.8 | 78.6 | 30.9 | 39.1 | 38.8 |
| | **DIVER** | 20.5 | 14.4 | 59.6 | 81.8 | 29.4 | 42.2 | 41.3 |
| | | *Train Temperature* | | | | | | |
| t = 1.0 | GPRO w/ Clip-Higher | 18.9 | 16.4 | 57.3 | 81.2 | 28.7 | 41.5 | 40.7 |
| | **DIVER** | 23.8 | 16.4 | 60.9 | 84.4 | 29.4 | 44.0 | 43.1 |
| t = 0.6 | GPRO w/ Clip-Higher | 18.0 | 14.1 | 54.9 | 80.0 | 29.8 | 40.3 | 39.5 |
| | **DIVER** | 20.6 | 14.4 | 59.8 | 83.2 | 33.1 | 42.6 | 42.8 |

# E CASE STUDY

## Question and Answer

**Prompt:** *<system prompt>* This is the problem: For $k \geq 3$, we define an ordered $k$-tuple of real numbers $(x_1, x_2, \ldots, x_k)$ to be special if, for every $i$ such that $1 \leq i \leq k$, the product $x_1 \cdot x_2 \cdot \ldots \cdot x_k = x_i^2$. Compute the smallest value of $k$ such that there are at least 2009 distinct special $k$-tuples.

**Answer:** "\$12\$"

## DIVER Case

### Response 1:

*<think>*
1. If $P = 1$:
$$x_1 \cdot x_2 \cdot \cdots \cdot x_k = 1.$$

Then for each $i$:
$$1 = x_i^2 \implies x_i = 1 \text{ or } x_i = -1.$$

Each $x_i$ can be either 1 or -1, and the product of all $x_i$ must be 1. We need to count the number of such $k$-tuples. The number of $k$-tuples of 1's and $-1$'s (i.e., each $x_i$ can be 1 or $-1$) whose product is 1 can be counted as follows: the number of $-1$'s must be even (since an even number of $-1$'s will result in a product of 1). The total number of $k$-tuples where each $x_i$ is either 1 or $-1$ is $2^k$. Half of these tuples will have an even number of $-1$'s (since changing the sign of any one element changes the parity of the number of $-1$'s), hence there are $2^{k-1}$ such tuples.
2. If $P = -1$:
$$x_1 \cdot x_2 \cdot \cdots \cdot x_k = -1.$$

Then for each $i$:
$$-1 = x_i^2$$

This is impossible for real numbers since $x_i^2 \geq 0$ for all real $x_i$. Thus, there are no solutions when $P = -1$ with real numbers.
Therefore, the valid solutions come only from the case $P = 1$, and there are $2^{k-1}$ such $k$-tuples. We need $2^{k-1} \geq 2009$:
$$2^{k-1} \geq 2009.$$

To find the smallest $k$:
$$k - 1 \geq \log_2 2009.$$

We calculate $\log_2 2009$:
$$\log_2 2009 \approx 10.97$$

Since $k - 1$ must be an integer, we take the ceiling of 10.97, thus:
$$k - 1 \geq 11 \implies k \geq 12.$$

*</think>*
From the analysis above, the smallest $k$ such that there are at least 2009 distinct special $k$-tuples is $k = 12$, because $2^{11} = 2048 \geq 2009$, and $k = 11$ would yield only $2^{10} = 1024$ tuples, which is less than 2009.
Hence, the smallest $k$ for which there are at least 2009 distinct special $k$-tuples is $k = 12$.
The final answer is:

$$\boxed{12}$$

------------------------------------------------------------

### Response 2:

*<think>*
For $P \neq 0$, we can divide both sides by $P$:

$$1 = P^{\frac{k}{2}-1} \cdot (-1)^{k-n}.$$

For this equation to hold, two conditions must be satisfied: 1. $P^{\frac{k}{2}-1} = 1$, implying $\frac{k}{2} - 1 = 0$ or $P = 1$ (because $P$ is a real number). 2. $(-1)^{k-n} = 1$, implying that $k - n$ must be even. From $\frac{k}{2} - 1 = 0$, we get:

$$\frac{k}{2} = 1 \implies k = 2.$$

However, the problem states $k \geq 3$, so $k = 2$ is not valid. Therefore, $P = 1$. So, $P = 1$ and thus $(\sqrt{P})^2 = 1$, meaning $x_i^2 = 1$ for all $i$. Therefore, each $x_i$ can only be 1 or $-1$. Now, we need $(-1)^{k-n} = 1$, meaning $k - n$ is even, implying $n$ and $k$ have the same parity (both are either even or both are odd). Next, let's determine the number of distinct $k$-tuples. If each $x_i$ can be either 1 or $-1$, there are $2^k$ possible $k$-tuples in total. However, not all of them will satisfy the condition, because they must maintain $P = 1$. The product $x_1 x_2 \cdots x_k = 1$ if and only if an even number of $x_i$'s are $-1$. The number of ways to choose an even number of $-1$'s out of $k$ positions is given by the sum of binomial coefficients of even indices:

$$\sum_{\substack{0 \leq j \leq k \\ j \text{ even}}} \binom{k}{j}.$$

It is known that the sum of binomial coefficients of even indices equals $2^{k-1}$:

$$\sum_{\substack{0 \leq j \leq k \\ j \text{ even}}} \binom{k}{j} = 2^{k-1}.$$

Thus, there are $2^{k-1}$ distinct special $k$-tuples. We need $2^{k-1} \geq 2009$:

$$2^{k-1} \geq 2009.$$

To find the smallest $k$:

$$k - 1 \geq \log_2(2009).$$

Since $\log_2(2009) \approx 10.97$, we round up to the next integer because $k-1$ must be an integer:

$$k - 1 \geq 11.$$

This gives $k \geq 12$. *</think>*
The smallest $k$ that satisfies $2^{k-1} \geq 2009$ is $k = 12$. To verify, when $k = 12$:

$$2^{12-1} = 2^{11} = 2048,$$

which is indeed greater than or equal to 2009. Thus, the smallest $k$ such that there are at least 2009 distinct special $k$-tuples is $k = 12$. Therefore, the smallest $k$ is:

$$\boxed{12}$$

## Entropy-RL Case

### Response 1:

*<think>*
the given problem states that $k \geq 3$, so we need to reconsider another approach since $k = 2$ isn't applicable directly for $k \geq 3$. So let's say $P = t^2$ where $t = \sqrt{P}$ and then we need

$t^2 = t^k$. If we let $t = \sqrt{P}$, we get

$$t^2 = t^k \implies t^2 - t^k = 0 \implies t^2(1 - t^{k-2}) = 0.$$

This implies that $t^2 = 0$ or $t^{k-2} = 1$. First, if $t^2 = 0$, then $t = 0$, and thus all the coordinates should be zero yielding once again the trivial solution, leading to the previous case.

For $t^{k-2} = 1$, we have two solutions:

1. $t = 1$ and 2. $t = -1$.

When $t = 1$, then $P = 1$, and each $x_i = \pm 1$, and when $t = -1$, $P = 1$, and again each $x_i = \pm 1$.

Hence, each coordinate can be either $1$ or $-1$, giving $2^k$ total combinations of $k$-tuples. Since we already have the trivial all-zero solution giving just one, we'll focus on the remaining ones, so the total number of solutions is $2^k - 1$. We need $2^k - 1 \geq 2009$. Let's calculate smallest $k$:

We need to find the smallest integer $k$ such that

$$2^k - 1 \geq 2009,$$

or

$$2^k \geq 2010.$$

Now let's calculate powers of 2:

$$2^{10} = 1024,$$
$$2^{11} = 2048.$$

It follows that the smallest value for which $2^k \geq 2010$ is when $k = 11$.

</think>

Now that we've reasoned through the problem, we conclude:

That would mean $k = 11$ would yield $2^{11} = 2048$, and then there'd be $2048 - 1 = 2047$ nonzero solutions plus 1 trivial solution leading to 2048 total tuples; thus $k = 11$ suffices because even $k = 10$ would not yield enough (as $2^{10} - 1 = 1023 < 2009$). So the final answer must be:

$$\boxed{11}$$

--------------------------------------------------------------------------------

## Response 2:

<think>

least 2009 distinct special $k$-tuples, we must have

$$2^k \geq 2009.$$

We need to find the smallest integer $k$ such that $2^k \geq 2009$.

Let's calculate the powers of 2:

$$2^{10} = 1024,$$
$$2^{11} = 2048.$$

We can see that $2^{10} = 1024 < 2009$ and $2^{11} = 2048 > 2009$. Thus, the smallest $k$ such that $2^k \geq 2009$ is 11.

</think>

The smallest value of $k$ such that there are at least 2009 distinct special $k$-tuples is

$$\boxed{11}$$

## GRPO w/ Clip-higher Case

### Response 1:

*<think>*
If $x = 0$, then $P = 0$, and all $x_i$ must be zero, which gives us one more solution.

So, for $k > 3$, we have $k - 1$ distinct solutions (the $k - 2$ roots of unity plus the solution $x = 0$). We need at least 2009 distinct special $k$-tuples. Therefore, we need $k - 1 \geq 2009$, which gives us $k \geq 2010$. The smallest value of $k$ that satisfies this inequality is $k = 2010$. Let's verify this. For $k = 2010$, we have $2010 - 1 = 2009$ distinct solutions from the roots of unity plus the solution $x = 0$, which gives us exactly 2009 distinct special $k$-tuples. Therefore, the smallest value of $k$ such that there are at least 2009 distinct special $k$-tuples is $k = 2010$. *</think>*

The smallest value of $k$ such that there are at least 2009 distinct special $k$-tuples is

$$\boxed{2010}$$

-------------------------------------------------------------------------------------

### Response 2:

*<think>*
To count the number of distinct special $k$-tuples, we need to consider the number of solutions to these equations. If $k - 2$ is even, we have at least 2 distinct tuples $(1, 1, \ldots, 1)$ and $(-1, -1, \ldots, -1)$. If $k - 2$ is odd, we can have more complex tuples involving $-1$.

To have at least 2009 distinct special $k$-tuples, we need to find the smallest $k$ such that the number of solutions is at least 2009. If $k - 2$ is even, we have at least 2 solutions. If $k - 2$ is odd, we can have more solutions, but we need to count them carefully. For $k - 2$ odd, we can have tuples with an even number of $-1$'s and the rest 1's, and permutations of $(x, -x, -1, -1, \ldots, -1)$.

The number of distinct tuples grows quickly with $k$, and we need to find the smallest $k$ such that the number of solutions is at least 2009. Let's calculate this step by step. For $k - 2$ odd, the number of distinct tuples is related to the number of ways to choose positions for $-1$ and the values of $x$ and $-x$.

Let's check the number of solutions for different $k$:

- For $k = 3$, we have $(1, 1, 1)$ and $(-1, -1, -1)$, so 2 solutions. - For $k = 4$, we have $(1, 1, 1, 1)$, $(-1, -1, -1, -1)$, and permutations of $(x, -x, -1, -1)$, so more than 2 solutions. - For $k = 5$, we have $(1, 1, 1, 1, 1)$, $(-1, -1, -1, -1, -1)$, and more solutions involving $-1$.

We can see that as $k$ increases, the number of solutions increases. We need to find the smallest $k$ such that the number of solutions is at least 2009.

Let's calculate the number of solutions for $k = 2m + 3$ (since $k - 2$ must be odd):

- For $k = 2m + 3$, we have at least the tuples with all 1's and all $-1$'s, plus permutations of $(x, -x, -1, -1, \ldots, -1)$. The number of such tuples is related to the number of ways to choose positions for $-1$ and the values of $x$ and $-x$.

The number of distinct tuples for $k = 2m + 3$ is at least $2 +$ number of ways to choose positions for $-1$ and $x, -x$. This grows quickly with $k$.

By trial and error, we can see that for $k = 2011$, we have $k - 2 = 2009$, which is odd, and we can have at least 2009 distinct tuples.
*</think>*

The smallest value of $k$ such that there are at least 2009 distinct special $k$-tuples is $k = 2011$. The final answer is:

$$\boxed{2011}$$

## Pass@k Training Case

### Response 1:

*<think>*
Since $P = P^{k/2}$, we have two cases to consider: 1. If $P = 0$: In this case, all $x_i$ must be zero because otherwise, if any $x_i$ is non-zero, the product would not be zero. This gives us the $k$-tuple $(0, 0, \ldots, 0)$, which is indeed a solution. For $P = 0$, it forms one special $k$-tuple. 2. If $P \neq 0$: We can divide both sides of the equation by $P^{k/2}$ (since $P > 0$):

$$1 = (-1)^{k-a}.$$

This implies that $(-1)^{k-a} = 1$, meaning $k - a$ must be even. Consequently, $a$ and $k$ must have the same parity (both even or both odd). Moreover, since $P = (\sqrt{P})^2 = (\sqrt{P})^k \cdot (-1)^{k-a}$, we get:

$$1 = (-1)^{k-a}.$$

This implies $k - a$ is even, so $a$ and $k$ are of the same parity (both even or both odd).
Next, we need to count the number of distinct $k$-tuples. For non-zero $P$, the number of distinct $k$-tuples is the number of ways to choose $a$ (with $0 \leq a \leq k$) such that $a$ and $k$ share the same parity. The number of choices for $a$ (with $0 \leq a \leq k$) having the same parity as $k$ is $\frac{k+1}{2}$ if $k$ is odd and $\frac{k}{2} + 1$ if $k$ is even. Including the zero product case, we need the total number of distinct $k$-tuples to be at least 2009:

$$1 + (\text{number of } k\text{-tuples with } P \neq 0).$$

If $k$ is odd, there are $\frac{k+1}{2}$ choices for $a$, making the total number of distinct $k$-tuples:

$$1 + \frac{k+1}{2}.$$

If $k$ is even, there are $\frac{k}{2} + 1$ choices for $a$, making the total number of distinct $k$-tuples:

$$1 + \left(\frac{k}{2} + 1\right) = 1 + \frac{k}{2} + 1 = \frac{k}{2} + 2.$$

We need this total to be at least 2009. We will check both cases separately. 1. For $k$ odd: $1 + \frac{k+1}{2} \geq 2009$

$$\frac{k+1}{2} + 1 \geq 2009,$$

$$\frac{k+1}{2} \geq 2008,$$

$$k + 1 \geq 4016,$$

$$k \geq 4015.$$

2. For $k$ even: $\frac{k}{2} + 2 \geq 2009$

$$\frac{k}{2} + 2 \geq 2009,$$

$$\frac{k}{2} \geq 2007,$$

$$k \geq 4014.$$

The smallest $k$ that satisfies either condition is $k = 4015$ (since 4015 is odd and satisfies the first condition). Hence, the smallest $k$ is 4015.
*</think>*
To find the smallest $k$ such that there are at least 2009 distinct special $k$-tuples, we need to satisfy the condition based on the derived inequalities. We found that for $k \geq 4015$ (when $k$ is odd) or $k \geq 4014$ (when $k$ is even):
1. If $k$ is odd, the smallest $k$ satisfying $k \geq 4015$. 2. If $k$ is even, the smallest $k$ satisfying $k \geq 4014$.

Since 4015 is the smallest value from these two cases, it gives us the final answer:

$$\boxed{4015}$$

--------------------------------------------------------------------------------

## Response 2:

*<think>*
Now, let's consider the possible values of $x_i$. Since $P = x_i^2$, and $P$ is either 1 or $-1$ (if $k - 2$ is even), we have $x_i^2 = 1$ or $x_i^2 = -1$. The equation $x_i^2 = 1$ has two solutions: $x_i = 1$ or $x_i = -1$. The equation $x_i^2 = -1$ has no real solutions. Therefore, each $x_i$ must be either 1 or $-1$.

If $P = 1$, then the number of $-1$'s in the tuple must be even. If $P = -1$, then the number of $-1$'s in the tuple must be odd. We need to count the number of such tuples.

Let's denote the number of $-1$'s in the tuple by $m$. If $P = 1$, then $m$ must be even, and if $P = -1$, then $m$ must be odd. The number of ways to choose $m$ positions out of $k$ for the $-1$'s is $\binom{k}{m}$. So, the total number of special $k$-tuples is the sum of $\binom{k}{m}$ for all even $m$ plus the sum of $\binom{k}{m}$ for all odd $m$. It is known that the sum of the binomial coefficients for even $m$ and odd $m$ are equal and each sum to $2^{k-1}$. Therefore, the total number of special $k$-tuples is $2^{k-1} + 2^{k-1} = 2^k$.

We need this to be at least 2009. So, we need to find the smallest $k$ such that $2^k \geq 2009$. We can calculate the powers of 2:

$$2^{10} = 1024,$$
$$2^{11} = 2048.$$

Since $2048 \geq 2009$, the smallest $k$ is 11. We can verify that $2^{11} = 2048$, which is indeed greater than or equal to 2009.

Therefore, the smallest value of $k$ such that there are at least 2009 distinct special $k$-tuples is 11.
*</think>*

The smallest value of $k$ such that there are at least 2009 distinct special $k$-tuples is

$$\boxed{11}$$

## F   THE USE OF LARGE LANGUAGE MODELS (LLMS)

We utilize Large Language Models (LLMs) to assist with polishing the writing and improving text readability. Specifically, LLMs are employed for proofreading, enhancing grammar, and refining sentence structure. The LLM was used solely for editorial purposes to improve clarity and did not contribute to research ideation, experimental design, implementation, analysis, or scientific conclusions. All core research contributions, experiments, and analyses were conducted independently by the authors without LLM assistance.

