# OpenReview forum: "Diversity-Incentivized Exploration for Versatile Reasoning"
_ICLR.cc/2026/Conference — ICLR 2026 Poster_

### Official Review · Reviewer_BRzv · 2025-10-24

**Soundness:** 2
**Presentation:** 3
**Contribution:** 3
**Rating:** 6
**Confidence:** 2

**Summary:**

This paper studies RL from verifiable rewards for LLM reasoning. The authors argue that token-level exploration is not enough, and we should explore entire reasoning paths. They propose a small change to GRPO: for each prompt, sample G rollouts, measure sequence-level diversity across that group, and add this as an intrinsic reward. The bonus is only given when a rollout is verified correct and is clipped. On math benchmarks (AIME/AMC/Minerva/OlympiadBench/MATH-500) the method improves Pass@1/Pass@k. Overall, the contribution is mixed: conceptual (global/trajectory exploration) and algorithmic (a training-time diversity bonus) with solid empirical support in the math setting.

**Strengths:**

- Very simple, plug-and-play idea. Easy to add to GRPO and existing RL-LLM training frameworks.
- Clear empirical gains on hard math tasks and strong improvements.
- Training diagnostics are helpful (diversity/entropy/“solve-none”).
- The correctness mask and clipping are practical and effective to avoid length explosion.
- Works across several backbones and seems to be applicable to various LLM architectures.

**Weaknesses:**

- Theory vs practice mismatch. The shaping argument (policy invariance) assumes a state-only potential. Here the bonus is group-dependent and masked by correctness. The paper does not provide a formal justification for invariance in this setting.
- Equational diversity extraction is under-specified. The paper defines the set metric but does not describe the extraction pipeline (parser/regex, failures, error rate). Hard to reproduce precisely.
- The paper lacks ablation studies to understand the impact of the hyperparameters or the performance of the method.
- Scope limited to single-turn math. The “versatile reasoning” claim is not tested on code, multi-turn, or tool-use tasks.
- Code is not available which hinders reproducibility.

**Questions:**

- How exactly do you extract equations for ED (parser, regex, normalization)? What is the failure rate?
- What is the training overhead (tokens/sec, wall-clock) vs GRPO?
- How sensitive is the method to λ, group size G, and temperature?
- Could you make the code available for reproducibility?

---

> ### Author Response · Authors · 2025-11-21
> **Author Response (Part 1/2)**
>
> ***
>
> **`Q1. Theory vs practice mismatch. The shaping argument (policy invariance) assumes a state-only potential. Here the bonus is group-dependent and masked by correctness. The paper does not provide a formal justification for invariance in this setting.`**
>
> A1. **A shaping mechanism based on a state-only potential is fully compatible with the practice of computing bonuses within a group of responses**. The ingenuity of our reward-shaping design lies in that by setting the shaping reward as the difference between the diversities of adjacent states ($R_{\text{int}}(s_t,a_t,s_{t+1})=\gamma d(s_{t+1})-d(s_t)$ in Eq. (6)), the intrinsic reward for a complete query-response pair is derived as the diversity of the final response within the group ($R_{\text{int}}([q,o_i])=\sum_{t=0}^{T-1}\gamma^tR([q,o_{i,\le t}],o_{i,t+1},[q,o_{i,\le t+1}])=\gamma^Td([q,o_i])$ in Eq. (8)). **This elegant formulation avoids the need to calculate the diversity of any intermediate sentences**.
>
> Then, let’s look at the augmented reward function as $R'([q,o_i]) = R([q,o_i]) + \lambda\cdot R_{\text{int}}([q,o_i])$ in Eq. (9). The GRPO pipeline is: i) sampling a group of responses $\{o_i\}_{i=1}^G$, ii) calculating the group relative advantages $A_i$  from the reward $R([q,o_i])$, and iii) performing policy gradient updates. The original reward $R([q,o_i])$ is a state-only function, and calculating the advantage $A_i$ is group-dependent. When calculating the state-only and group-dependent advantages in the second step, we can naturally calculate each response’s diversity $d([q,o_i])$ within the group as the shaping reward (masked by the response’s correctness $R([q,o_i])$). Hence, **the practice of implementing DIVER’s reward shaping aligns precisely with the foundational assumptions of the theory**.
>
> In our manuscript, Eq. (8) (L262, p5) presents the practice of deriving the shaping reward, and Theorem 1 (L286, P6) provides the formal justification for the optimal policy invariance in our reward shaping setting.
>
> ***
>
> **`Q2. Equational diversity extraction is under-specified. The paper defines the set metric but does not describe the extraction pipeline (parser/regex, failures, error rate). Hard to reproduce precisely. How exactly do you extract equations for ED (parser, regex, normalization)? What is the failure rate?`**
>
> A2. We extract mathematical formulas using regular expressions that identify three common LaTeX notation patterns: `\\[ \\], \\( \\), and \$ \$`. Our implementation uses re.findall() to capture these patterns and collects them in a set to eliminate duplicates.
>
> For math tasks with standard mathematical notations, our method demonstrates high extraction accuracy. During equation extraction, it relies on human annotations to obtain the exact ground-truth count of equations in a response. For a demo illustration, we randomly sample 5 responses from different query inputs. The following table presents the equation counts via our extraction and corresponding ground-truth values, where the failure rate is about 8/95=8.4%. **Since ED serves as an intrinsic reward to encourage exploration, it does not need to be perfectly accurate; it only needs to approximate response diversity well enough to guide effective exploration**. Primary results in Table 1 (L325, P7) also demonstrate that DIVER-ED achieves an effective and broader exploration scope that unlocks enhanced reasoning capacity.
>
> |  | Response 1 | Response 2 | Response 3 | Response 4 | Response 5 |
> | --- | --- | --- | --- | --- | --- |
> | No. of equations via extraction | 14 | 25 | 13 | 17 | 18 |
> | ground-truth value | 17 | 25 | 18 | 17 | 18 |
>
> In summary, **DIVER attains high accuracy in equation extraction for computing the ED metric**, which has been shown to enhance reasoning performance. We will include the pipeline of extracting equations and the illustration of extraction accuracy in the revised manuscript. Also, we have fully documented these details in our source code uploaded to the supplementary material.

---

> ### Author Response · Authors · 2025-11-21
> **Author Response (Part 2/2)**
>
> ***
>
> **`Q3. The paper lacks ablation studies to understand the impact of the hyperparameters or the performance of the method. How sensitive is the method to  $\lambda$, group size G, and temperature?`**
>
> A3. As DIVER’s core is to formulate global diversity as an intrinsic reward to incentivize deep exploration, **we have conducted the ablation analysis on core components in the original manuscript,** such as the coefficient  $\lambda$ and reward hacking solutions.
>
> `Coefficient  $\lambda$`. Due to the limited space of the main paper, we included the analysis on the diversity coefficient $\lambda$ in Appendix B.2 (Figure 9, L836, P16). **Results demonstrate the robustness of DIVER’s reward shaping mechanism across different $\lambda$ values, making DIVER practical for applications without requiring precise tuning.** The ablation of reward hacking solutions is presented in Sec. 4.4 (L450, P9), where we thoroughly evaluated four different reward shaping strategies. As Figure 7 illustrates, **our conditional shaping mechanism effectively constrains exploration to valid solutions and mitigates reward hacking**. This analysis verifies the robustness of DIVER’s principled design of the reward shaping mechanism.
>
> `Reward hacking`. The ablation of reward hacking solutions is presented in Sec. 4.4 (L450, P9), where we thoroughly evaluated four different reward shaping strategies of applying diversity rewards to 1) correct responses, 2) all responses, 3) errors only, and 4) all responses with a length penalty. As Figure 7 illustrates, **our conditional shaping mechanism effectively constrains exploration to valid solutions and mitigates reward hacking**. This analysis verifies the robustness of DIVER’s principled design of the reward shaping mechanism.
>
> Regarding the group size and temperature, we maintained consistent settings (group size 8, training temperature 1.0, and evaluation temperature 0.6) across all baselines and DIVER to ensure a fair comparison. Due to the limited time for posting the first round of rebuttal, we did not have enough time to conduct more ablation studies to analyze the impact of these trivial hyperparameters. We will continue to conduct hyperparameter analysis on group size and temperature, and upload the results before the final discussion deadline Dec. 3rd.
>
> ***
>
> **`Q4. Scope limited to single-turn math. The “versatile reasoning” claim is not tested on code, multi-turn, or tool-use tasks.`**
>
> A4. Thank you for your inspiring comment on the limitation. Following the common practice in mainstream RLVR studies, we focus on mathematical reasoning as a rigorous testbed with verifiable outcomes. **Comprehensive experiments demonstrate promising results on in-domain tasks and stronger generalization on out-of-domain tasks (ARC-c, GPQA\*, and MMLU-Pro)**. We agree that it is an important direction to explore DIVER's effectiveness in more diverse reasoning contexts, such as the code, multi-turn, and tool-use tasks. We have included the above limitation and promising direction in the Conclusion section, and leave it as future work.
>
> ***
>
> **`Q5. Code is not available which hinders reproducibility. Could you make the code available for reproducibility?`**
>
> A5. Absolutely! We have provided our source code and the scripts with all hyperparameter configurations in the supplementary material. We would be happy to engage in further discussion if you have any questions about our code.
>
> ***
>
> **`Q6. What is the training overhead (tokens/sec, wall-clock) vs GRPO?`**
>
> A6. Compared to GRPO, DIVER’s increase in training overhead mainly comes from the diversity calculation. **The computation overhead of calculating textual diversity and equational diversity is negligible**, since both metrics only involve rule-based calculations (e.g., n-gram matching for BLEU score or string recognition for equation extraction) without any feedforward or backpropagation of large-scale models. Following your advice, we have included a quantified analysis of the computation cost. The following result shows that DIVER only incurs a 5% increase in training time compared to GRPO and Entropy-RL baselines, requiring less training time than Pass@k Training baseline. The empirical evidence matches the above intuition of computation overhead. We will include the computational cost analysis in the revised paper.
>
> |  | DIVER(ours) | GRPO w/ Clip-Higher | Entropy-RL | Pass@k Training |
> | --- | --- | --- | --- | --- |
> | **Time Cost (350 Steps)** | 29.75h | 28.46h | 28.13h | 30.73h |

---

> ### Author Response · Authors · 2025-11-21
> **Author Summary Response**
>
> Thank you for your valuable review comments, which help us gain more critical insights and significantly enhance our empirical evaluation. **We are honored to have your positive comments on the motivation** (”very simple, plug-and-play idea”), **the method** (”easy to add to existing frameworks…the correctness mask and clipping are practical and effective”), **and the empirical evaluation** (”clear empirical gains…training diagnostics are helpful”).
>
> Also, we are grateful that **other reviewers** (we refer to wris as R1, mNMY as R2, and RcXm as R3) **make positive comments on our overall contributions, including the motivation** (”studies an important problem” by R1, ”presents a clear motivation” by R2, and “the motivation…is clear and compelling” by R3)**, the method** (”a well-structured method” by R2 and “the method is simple and pricipled” by R3)**, and the empirical evaluation with superior performance** (”comprehensive experimental results…DIVER achieves performance improvement across different benchmarks” by R1, ”experiments are comprehensive and demonstrate consistent improvements” by R2, and “experimental evaluation is thorough” by R3). We hope these collective evaluations could help clarify the breadth and strength of our contributions.
>
> **We have made a number of justifications and extended experimental analyses to address your concerns**. Due to limited time for posting the first round of rebuttal, **we will include more experimental results and update the paper thoroughly before the final discussion deadline Dec. 3rd**. Please let us know if we have addressed your concerns. We are more than delighted to have further discussions and improve our manuscript.

---

### Official Review · Reviewer_RcXm · 2025-10-28

**Soundness:** 3
**Presentation:** 3
**Contribution:** 3
**Rating:** 6
**Confidence:** 4

**Summary:**

The paper identifies global sequence-level diversity as a driver for better exploration and improved reasoning capability in RL finetuning of LLMs, and proposes DIVER: an intrinsic-reward framework that (1) measures sequence diversity (Textual Diversity via inverted BLEU, and Equational Diversity via formula uniqueness), (2) injects diversity as a potential-based shaping reward to preserve optimal policy invariance, and (3) uses simple heuristics to mitigate reward-hacking. Experiments on several math benchmarks and general domains show consistent gains over GRPO and other exploration baselines; ablations and diagnostics support the claims.

**Strengths:**

1. The motivation to enhance exploration through global sequence-level diversity is clear and compelling.
2. The method is simple and pricipled.
3. The experimental evaluation is thorough.

**Weaknesses:**

1. Limited generality of diversity metrics: The choice of diversity metrics lacks clear guidance. The two proposed ones have different strengths and weaknesses and may not generalize to all scenarios. It would be natural to consider combining them or testing alternatives such as LLM-based or embedding-based measures.
2. No analysis of computational cost: The paper does not discuss whether computing diversity metrics adds significant overhead or affects training efficiency.
3. Theorem 1 offers limited novelty: It follows directly from prior reward-shaping theory[1], and the reward-hacking solution breaks its assumptions, reducing its relevance.
4. Insufficient ablations and sensitivity analysis: Key components and hyperparameters (e.g., λ, reward-hacking solution) are not thoroughly tested for robustness.
5. Missing discussion of potentially related prompt selection methods: Prior work promotes uncertainty by selecting prompts with success rates near 0.5 [2,3,4]. Since both pursue response uncertainty, their connection deserves discussion.

[1] Policy invariance under reward transformations: Theory and application to reward shaping.\
[2] Self-Evolving Curriculum for LLM Reasoning\
[3] Online Difficulty Filtering for Reasoning Oriented Reinforcement Learning\
[4] Can prompt difficulty be online predicted for accelerating rl finetuning of reasoning models?

**Questions:**

see weaknesses

---

> ### Author Response · Authors · 2025-11-21
> **Author Response (Part 1/3)**
>
> ***
>
> **`Q1. Limited generality of diversity metrics: The choice of diversity metrics lacks clear guidance. The two proposed ones have different strengths and weaknesses and may not generalize to all scenarios. It would be natural to consider combining them or testing alternatives such as LLM-based or embedding-based measures.`**
>
> A1. `Alternative measures`. To demonstrate the generality of our diversity metrics, we conduct a correlation analysis comparing TD and ED with alternative diversity measures: 1) **LLM-based diversity**: we compute the cosine similarity of the model's hidden states from the final layer outputs; 2) **Embedding-based diversity**: we use `google/embeddinggemma-300m` to encode responses into embeddings and compute their cosine similarity.
>
> For each method (DIVER and baselines), we generate multiple responses per prompt and compute all diversity metrics within each response group. Evaluated on 1,560 rollouts total, the average metrics are shown in the table (↑ indicates higher values represent greater diversity; ↓ indicates the opposite). The results demonstrate strong consistency between TD/ED and LLM-/embedding-based alternatives. **Methods achieving higher TD/ED scores consistently exhibit lower LLM-/embedding-based semantic similarity scores**, confirming that our choice of diversity metrics effectively captures meaningful semantic diversity rather than merely surface-level variation. **This validates the generality of TD and ED as reliable proxies for reasoning diversity**. We will include this insightful analysis on diversity metrics in the revised manuscript.
>
> | **Method** | **TD (↑)** | **ED (↑)** | **LLM-based (↓)** | **embedding-based (↓)** | pass@1 | pass@8 |
> | --- | --- | --- | --- | --- | --- | --- |
> | **DIVER(ours)** | 0.702 | 0.477 | 0.936 | 0.881 | 43.1 | 60.8 |
> | GRPO w/ Clip-Higher | 0.805 | 0.575 | 0.916 | 0.841 | 40.7 | 57.4 |
> | Entropy-RL | 0.549 | 0.294 | 0.947 | 0.954 | 41.8 | 58.0 |
> | Pass@k Training | 0.441 | 0.200 | 0.959 | 0.967 | 41.5 | 56.3 |
>
> Moreover, our diversity metrics are both computationally lightweight and easy to implement. **The computation overhead of calculating textual diversity and equational diversity is negligible**, since both metrics only involve rule-based calculations (e.g., n-gram matching for BLEU score or string recognition for equation extraction) without any feedforward or backpropagation of large-scale models (e.g., the above LLM-/embedding-based metrics).
>
> `Combining metrics`. Thank you for the insightful suggestion to combine our two diversity metrics. Following your recommendation, we implemented a mixed metric (DIVER-MIX) that combines TD and ED. Superingly, **DIVER-MIX achieves even better performance than using either metric alone!** As shown below, DIVER-MIX outperforms both DIVER-TD and DIVER-ED variants, achieving the highest average accuracy on both in-domain (43.1%) and out-of-domain (58.8%) tasks. We will adopt your valuable suggestion to further improve DIVER’s design in the revised paper.
>
> | **Method** | **In-Domain** |  |  |  |  |  | **Out-of-Domain** |  |  |  |
> | --- | --- | --- | --- | --- | --- | --- | --- | --- | --- | --- |
> |  | **AIME24/25** | **AMC** | **MATH-500** | **Minerva** | **Olympiad** | **Avg.** | **ARC-c** | **GPQA*** | **MMLU-Pro** | **Avg.** |
> | **GPRO w/ Clip-Higher** | 18.9/16.4 | 57.3 | 81.2 | 28.7 | 41.5 | 40.7 | 82.1 | 36.2 | 47.2 | 55.2 |
> | **Entropy-RL** | 23.6/12.8 | 58.4 | 82.8 | 31.6 | 41.5 | 41.8 | 80.7 | 38.8 | 48.4 | 56.0 |
> | **Pass@k Traning** | 20.9/15.7 | 52.3 | 83.8 | **32.7** | 43.8 | 41.5 | 79.3 | 37.8 | 49.0 | 55.3 |
> | **DVIER-TD** | 22.5/16.9 | 59.4 | 82.2 | 27.9 | 44.7 | 42.3 | 83.4 | **42.3** | 49.5 | 58.4 |
> | **DVIER-ED** | 20.9/15.7 | 59.7 | 84.0 | 31.6 | **46.1** | 43.0 | 83.4 | 36.2 | 49.9 | 56.5 |
> | **DVIER-MIX** | 23.8/16.4 | **60.9** | **84.4** | 29.4 | 44.0 | **43.1** | **84.1** | 41.3 | **51.0** | **58.8** |
>
> ***
>
> **`Q2. No analysis of computational cost: The paper does not discuss whether computing diversity metrics adds significant overhead or affects training efficiency.`**
>
> A2. **The computation overhead of calculating textual diversity and equational diversity is negligible**, since both metrics only involve rule-based calculations (e.g., n-gram matching for BLEU score or string recognition for equation extraction) without any feedforward or backpropagation of large-scale models. Following your advice, we have included a quantified analysis of the computation cost. The following result shows that DIVER only incurs a 5% increase in training time compared to GRPO and Entropy-RL baselines, requiring less training time than Pass@k Training baseline. The empirical evidence matches the above intuition of computation overhead. We will include the computational cost analysis in the revised paper.
>
> |  | DIVER(ours) | GRPO w/ Clip-Higher | Entropy-RL | Pass@k Training |
> | --- | --- | --- | --- | --- |
> | **Time Cost (350 Steps)** | 29.75h | 28.46h | 28.13h | 30.73h |

---

> ### Author Response · Authors · 2025-11-21
> **Author Response (Part 2/3)**
>
> ***
>
> **`Q3. Insufficient ablations and sensitivity analysis: Key components and hyperparameters (e.g., $\lambda$, reward-hacking solution) are not thoroughly tested for robustness.`**
>
> A3. As DIVER’s core is to formulate global diversity as an intrinsic reward to incentivize deep exploration**, we have conducted ablation studies on key components in the original manuscript, such as the diversity coefficient $\lambda$ and reward hacking solutions**.
>
> `Coefficient $\lambda$`. Due to space limitation in the main paper, we included the sensitivity analysis of coefficient $\lambda$ in Appendix B.2 (L836, p16). As Figure 9 demonstrates, **DIVER exhibits remarkable robustness across different $\lambda$ values**. Surprisingly, even a small coefficient ($\lambda=0.001$) can provide sufficient diversity incentive. The possible reason is that, as the original reward $R$ is binary, a small scale of the intrinsic reward $R_{\text{int}}$ can already influence the landscape of the reward distribution. Then, increasing the value ($\lambda=0.01/0.1$) shows only minor performance variations without triggering reward hacking behaviors. Finally, with a large value $\lambda=1$, DIVER’s performance degrades a little. When $\lambda = 0$, the method reduces to baseline GRPO, as shown in Table 1 (L325, P7). **This hyperparameter analysis demonstrates the robustness of our reward shaping mechanism, making DIVER practical for applications without requiring precise tuning.**
>
> `Reward hacking`. The ablation of reward hacking solutions is presented in Sec. 4.4 (L450, P9), where we thoroughly evaluated four different reward shaping strategies of applying diversity rewards to 1) correct responses, 2) all responses, 3) errors only, and 4) all responses with a length penalty. As Figure 7 illustrates, **our conditional shaping mechanism effectively constrains exploration to valid solutions and mitigates reward hacking**. This analysis verifies the robustness of DIVER’s principled design of the reward shaping mechanism.
>
> Due to the limited time for posting the first round of rebuttal, we did not have enough time to conduct more ablation studies to analyze the impact of other trivial hyperparameters, such as the group size and temperature. We will continue to conduct more ablation studies to further demonstrate DIVER’s effectiveness, and upload the results before the final discussion deadline Dec. 3rd.
>
> ***
>
> **`Q4. Missing discussion of potentially related prompt selection methods: Prior work promotes uncertainty by selecting prompts with success rates near 0.5 [2,3,4]. Since both pursue response uncertainty, their connection deserves discussion.`**
>
> A4. Prompt selection methods [2-4] are data-level strategies that improve training efficiency by selecting samples of appropriate difficulty. In contrast, DIVER represents an algorithmic-level approach that changes the model's learning objectives through reward shaping, encouraging deeper exploration. Methodologically, prompt selection approaches are more similar to our illustrative experiment in Figure 3 (Sec. 3.2, L162, P4):  improving exploration efficiency by filtering out more diverse rollouts, which are all efforts at the data level. **DIVER and these data-end methods are not mutually exclusive and can complement each other in practice**. We appreciate your thoughtful consideration! Such discussions help provide a more comprehensive understanding of our method. We will incorporate these extended references and corresponding insights into our related work section in the revised paper.
>
> [2] Chen et al., Self-evolving curriculum for LLM reasoning, arXiv:2505.14970, 2025.
>
> [3] Bae et al., Online difficulty filtering for reasoning oriented RL, arXiv:2504.03380, 2025.
>
> [4] Qu et al., Can prompt difficulty be online predicted for accelerating RL finetuning of reasoning models? arXiv:2507.04632, 2025.

---

> ### Author Response · Authors · 2025-11-21
> **Author Response (Part 3/3) and Summary Response**
>
> ***
>
> **`Q5. Insufficient ablations and sensitivity analysis: Key components and hyperparameters (e.g., $\lambda$, reward-hacking solution) are not thoroughly tested for robustness.`**
>
> A5. As DIVER’s core is to formulate global diversity as an intrinsic reward to incentivize deep exploration**, we have conducted ablation studies on key components in the original manuscript, such as the diversity coefficient $\lambda$ and reward hacking solutions**.
>
> `Coefficient $\lambda$`. Due to space limitation in the main paper, we included the sensitivity analysis of coefficient $\lambda$ in Appendix B.2 (L836, p16). As Figure 9 demonstrates, **DIVER exhibits remarkable robustness across different $\lambda$ values**. Surprisingly, even a small coefficient ($\lambda=0.001$) can provide sufficient diversity incentive. The possible reason is that, as the original reward $R$ is binary, a small scale of the intrinsic reward $R_{\text{int}}$ can already influence the landscape of the reward distribution. Then, increasing the value ($\lambda=0.01/0.1$) shows only minor performance variations without triggering reward hacking behaviors. Finally, with a large value $\lambda=1$, DIVER’s performance degrades a little. When $\lambda = 0$, the method reduces to baseline GRPO, as shown in Table 1 (L325, P7). **This hyperparameter analysis demonstrates the robustness of our reward shaping mechanism, making DIVER practical for applications without requiring precise tuning.**
>
> `Reward hacking`. The ablation of reward hacking solutions is presented in Sec. 4.4 (L450, P9), where we thoroughly evaluated four different reward shaping strategies of applying diversity rewards to 1) correct responses, 2) all responses, 3) errors only, and 4) all responses with a length penalty. As Figure 7 illustrates, **our conditional shaping mechanism effectively constrains exploration to valid solutions and mitigates reward hacking**. This analysis verifies the robustness of DIVER’s principled design of the reward shaping mechanism.
>
> Due to the limited time for posting the first round of rebuttal, we did not have enough time to conduct more ablation studies to analyze the impact of other trivial hyperparameters, such as the group size and temperature. We will continue to conduct more ablation studies to further demonstrate DIVER’s effectiveness, and upload the results before the final discussion deadline Dec. 3rd.
>
> ### **`Summary Response`**
>
> Thank you for your valuable review comments, which help us gain more critical insights and significantly enhance our empirical evaluation. **We are honored to have your positive comments on the motivation** (”motivation…is clear and compelling”), **the method** (”method is simple and principled”), **and the empirical evaluation** (”experimental evaluation is thorough”).
>
> Also, we are grateful that **other reviewers** (we refer to wris as R1, mNMY as R2, and BRzv as R4) **make positive comments on our overall contributions, including the motivation** (”studies an important problem” by R1, ”presents a clear motivation” by R2, and “very simple, plug-and-play idea” by R4)**, the method** (”a well-structured method” by R2 and “easy to add to existing frameworks…the correctness mask and clipping are practical and effective” by R4)**, and the empirical evaluation with superior performance** (”comprehensive experimental results…DIVER achieves performance improvement across different benchmarks” by R1, ”experiments are comprehensive and demonstrate consistent improvements” by R2, and “clear empirical gains…training diagnostics are helpful” by R4). We hope these collective evaluations could help clarify the breadth and strength of our contributions.
>
> **We have made a number of justifications and extended experimental analyses to address your concerns**. Due to limited time for posting the first round of rebuttal, **we will include more experimental results and update the paper thoroughly before the final discussion deadline Dec. 3rd**. Please let us know if we have addressed your concerns. We are more than delighted to have further discussions and improve our manuscript.

---

### Official Review · Reviewer_mNMY · 2025-10-31

**Soundness:** 3
**Presentation:** 3
**Contribution:** 2
**Rating:** 4
**Confidence:** 4

**Summary:**

The paper proposes DIVER, a reinforcement learning framework that enhances reasoning in LLMs by introducing global sequence-level diversity as an intrinsic reward.Built upon GRPO, DIVER uses a potential-based reward shaping to preserve optimal policy invariance and mitigate reward hacking.Experiments on multiple math reasoning benchmarks show consistent but modest gains in Pass@1 and Pass@k (up to 32) compared to Entropy-RL and Pass@k Training.

**Strengths:**

1. The paper presents a clear motivation and a well-structured method for enhancing exploration in LLM reasoning.

2. Experiments are comprehensive and demonstrate consistent improvements across multiple benchmarks and model scales.

**Weaknesses:**

1. The improvement on Pass@k is relatively small and only evaluated up to k = 32, while [1] tests up to k = 1024, including the base model.

2. The comparison on Pass@1 in Table 1 seems not entirely fair, since methods like Pass@k Training [2] mainly aim to improve Pass@k. Their Pass@1 performance is usually enhanced through a two-stage training process (first optimizing Pass@k, then fine-tuning for Pass@1).

3. In Table 3, GRPO mostly remains the best baseline, and the base model results are not included, making DIVER’s advantage less consistent.

4. The diversity metrics (BLEU, formula overlap) may not effectively capture semantic-level reasoning diversity.

**Questions:**

1. One confusing point is that in Table 3, GRPO is actually the best baseline in most cases (compared to Entropy-RL and Pass@k Training), which seems inconsistent with the expected performance of these methods.

2. Would DIVER still outperform GRPO and the base model at larger values of k?

[1] Yang Yue, Zhiqi Chen, Rui Lu, Andrew Zhao, Zhaokai Wang, Shiji Song, and Gao Huang. Does Reinforcement Learning Really Incentivize Reasoning Capacity in LLMs Beyond the Base Model? arXiv preprint arXiv:2504.13837, 2025.

[2] Zhipeng Chen, Xiaobo Qin, Youbin Wu, Yue Ling, Qinghao Ye, Wayne Xin Zhao, and Guang Shi. Pass@K Training for Adaptively Balancing Exploration and Exploitation of Large Reasoning Models. arXiv preprint arXiv:2508.10751, 2025.

---

> ### Author Response · Authors · 2025-11-21
> **Author Response (Part 1/3)**
>
> ***
>
> **`Q1. The improvement on Pass@k is relatively small and only evaluated up to k=32, while [1] tests up to k = 1024, including the base model. Would DIVER still outperform GRPO and the base model at larger values of k?`**
>
> A1. Following your advice, we extend our experiments to $k=128, 256, 512, \text{and}~1024$ across AIME24, AIME25, and AMC benchmarks, including the base model Qwen2.5-Math-7B and all baselines. The results demonstrate **consistent superiority of DIVER across all settings**: 1) **DIVER achieves the highest Pass@k scores** across nearly all benchmarks and $k$ values, reaching 86.7% on AIME24 and 100% on AMC at Pass@1024. 2) **The performance gap increases with a larger $k$**, highlighting DIVER's superior ability to generate diverse, correct solutions. On AIME24, DIVER outperforms the second-best baseline by 6.7 points at Pass@1024. 3) **All RL methods substantially improve over the base model**, but DIVER consistently achieves the best results, confirming that diversity-driven exploration provides significant advantages even at large sample sizes. We will include these comprehensive results in the revised paper.
>
> | **Method** | **Benchmark** | **pass@128** | **pass@256** | **pass@512** | **pass@1024** |
> | --- | --- | --- | --- | --- | --- |
> | **Qwen2.5-Math-7B** | AIME24 | 63.3 | 70.0 | 76.7 | 80.0 |
> |  | AIME25 | 40.0 | 56.7 | 63.3 | 66.7 |
> |  | AMC | 92.8 | **97.6** | 97.6 | 97.6 |
> | **GPRO w/ Clip-Higher** | AIME24 | 70.0 | 73.3 | 76.7 | 76.7 |
> |  | AIME25 | 60.0 | 63.3 | 66.7 | 70.0 |
> |  | AMC | 94.0 | 96.4 | 97.6 | 98.8 |
> | **Entropy-RL** | AIME24 | 70.0 | 73.3 | 76.7 | 76.7 |
> |  | AIME25 | 63.3 | 63.3 | 66.7 | 73.3 |
> |  | AMC | **96.4** | 96.4 | 97.6 | 98.8 |
> | **Pass@k Traning** | AIME24 | 70.0 | 76.7 | 80.0 | 80.0 |
> |  | AIME25 | 63.3 | 63.3 | 70.0 | 73.3 |
> |  | AMC | 95.2 | 96.4 | 97.6 | **100.0** |
> | **DVIER(ours)** | AIME24 | 76.7 | 80.0 | 83.3 | 86.7 |
> |  | AIME25 | 63.3 | 66.7 | 70.0 | 76.7 |
> |  | AMC | **96.4** | **97.6** | **98.8** | **100.0** |
>
> ***
>
> **`Q2. The comparison on Pass@1 in Table 1 seems not entirely fair, since methods like Pass@k Training [2] mainly aim to improve Pass@k. Their Pass@1 performance is usually enhanced through a two-stage training process (first optimizing Pass@k, then fine-tuning for Pass@1).`**
>
> A2. To ensure a fair comparison with Pass@k Training, we re-ran experiments using a two-stage training process: optimizing Pass@k for the first 150 steps, then fine-tuning on Pass@1 for the subsequent 200 steps. The results show that **Pass@k Training improves on in-domain benchmarks** after two-stage training, with the average in-domain performance increasing from 39.9% to 41.5%. However, **DIVER variants still consistently outperform Pass@k Training across both domains**. DIVER-ED achieves 43.0% on in-domain tasks and DIVER-TD achieves 58.4% on out-of-domain tasks, surpassing the two-stage Pass@k Training by 1.5 and 3.1 points, respectively. We will include the comparison results in this new setting in the revised paper.
>
> | **Method** | **In-Domain** |  |  |  |  |  | **Out-of-Domain** |  |  |  |
> | --- | --- | --- | --- | --- | --- | --- | --- | --- | --- | --- |
> |  | **AIME24/25** | **AMC** | **MATH-500** | **Minerva** | **Olympiad** | **Avg.** | **ARC-c** | **GPQA*** | **MMLU-Pro** | **Avg.** |
> | **GPRO w/ Clip-Higher** | 18.9/16.4 | 57.3 | 81.2 | 28.7 | 41.5 | 40.7 | 82.1 | 36.2 | 47.2 | 55.2 |
> | **Entropy-RL** | **23.6**/12.8 | 58.4 | 82.8 | 31.6 | 41.5 | 41.8 | 80.7 | 38.8 | 48.4 | 56.0 |
> | **Pass@k Traning** | 20.9/15.7 | 52.3 | 83.8 | **32.7** | 43.8 | 41.5 | 79.3 | 37.8 | 49.0 | 55.3 |
> | **DVIER-TD** | 22.5/**16.9** | 59.4 | 82.2 | 27.9 | 44.7 | 42.3 | **83.4** | **42.3** | 49.5 | **58.4** |
> | **DVIER-ED** | 20.9/15.7 | **59.7** | **84.0** | 31.6 | **46.1** | **43.0** | **83.4** | 36.2 | **49.9** | 56.5 |

---

> ### Author Response · Authors · 2025-11-21
> **Author Response (Part 2/3)**
>
> ***
>
> **`Q3. In Table 3, GRPO mostly remains the best baseline, and the base model results are not included, making DIVER’s advantage less consistent. One confusing point is that in Table 3, GRPO is actually the best baseline in most cases (compared to Entropy-RL and Pass@k Training), which seems inconsistent with the expected performance of these methods.`**
>
> A3. We clarify that the GRPO baseline uses the **clip-higher technique from DAPO** [1], not vanilla GRPO, which explains its strong performance. To rigorously demonstrate DIVER's consistent advantage, we **repeat all experiments across Pass@2 to Pass@32,** and report the means and standard deviations for a total of 3 runs. The results show that **DIVER consistently outperforms all baselines across all $k$ values**, validating its consistent superiority over exploration-based baselines.
>
> | **Pass@k** | **Model** | **AIME 24** | **AIME 25** | **AMC** | **MATH-500** | **Minerva** | **Olympiad** | **Avg.** |
> | --- | --- | --- | --- | --- | --- | --- | --- | --- |
> | **k = 2** | Qwen2.5-Math-7B | 23.3 ± 3.3 | 12.2 ± 5.1 | 55.4 ± 1.2 | 74.7 ± 0.8 | 27.2 ± 1.3 | 35.5 ± 0.8 | 38.1 ± 0.3 |
> |  | GPRO w/ Clip-Higher | 26.6 ± 5.8 | 18.9 ± 1.9 | 61.4 ± 3.7 | 85.7 ± 1.5 | 38.0 ± 0.2 | 50.4 ± 0.7 | 46.8 ± 1.3 |
> |  | Entropy-RL | 26.7 ± 0.0 | 22.2 ± 3.9 | 63.9 ± 1.2 | 86.3 ± 0.5 | 35.5 ± 0.8 | 50.5 ± 2.4 | 47.5 ± 0.7 |
> |  | Pass@k Training | 23.3 ± 3.4 | 22.2 ± 6.9 | 63.5 ± 0.7 | 85.1 ± 1.0 | 40.7 ± 1.2 | 47.6 ± 1.6 | 47.1 ± 1.7 |
> |  | **DIVER(ours)** | 28.9 ± 5.1 | 27.8 ± 8.4 | 67.9 ± 1.4 | 87.7 ± 0.1 | 42.9 ± 1.1 | 52.1 ± 1.5 | **51.2 ± 1.9** |
> | **k = 4** | Qwen2.5-Math-7B | 30.0 ± 5.8 | 16.6 ± 5.8 | 67.1 ± 3.5 | 81.5 ± 0.6 | 35.5 ± 0.6 | 43.8 ± 0.5 | 45.8 ± 1.7 |
> |  | GPRO w/ Clip-Higher | 32.2 ± 11.7 | 25.5 ± 3.9 | 72.3 ± 1.2 | 88.9 ± 1.1 | 47.3 ± 1.1 | 53.3 ± 0.9 | 53.2 ± 0.9 |
> |  | Entropy-RL | 38.9 ± 3.8 | 28.9 ± 3.8 | 74.3 ± 2.8 | 89.5 ± 0.1 | 40.6 ± 1.2 | 54.7 ± 0.8 | 54.5 ± 1.3 |
> |  | Pass@k Training | 36.7 ± 3.4 | 24.4 ± 8.4 | 71.5 ± 3.0 | 88.5 ± 0.2 | 42.2 ± 1.1 | 55.7 ± 0.7 | 53.2 ± 1.6 |
> |  | **DIVER(ours)** | 37.8 ± 1.9 | 24.4 ± 3.9 | 75.9 ± 4.3 | 90.5 ± 0.5 | 47.3 ± 1.9 | 57.7 ± 0.4 | **55.6 ± 1.6** |
> | **k = 8** | Qwen2.5-Math-7B | 42.2 ± 5.1 | 23.3 ± 3.4 | 77.1 ± 2.1 | 86.7 ± 1.2 | 41.3 ± 0.6 | 51.3 ± 0.9 | 53.7 ± 1.1 |
> |  | GPRO w/ Clip-Higher | 35.6 ± 5.1 | 30.0 ± 5.8 | 77.9 ± 1.8 | 90.7 ± 0.3 | 52.1 ± 1.3 | 58.5 ± 2.1 | 57.4 ± 1.1 |
> |  | Entropy-RL | 41.1 ± 1.9 | 28.9 ± 3.8 | 80.3 ± 2.5 | 91.9 ± 0.8 | 45.6 ± 1.0 | 59.9 ± 0.5 | 58.0 ± 1.2 |
> |  | Pass@k Training | 36.7 ± 3.4 | 24.4 ± 2.0 | 80.3 ± 0.7 | 90.7 ± 0.1 | 45.5 ± 1.9 | 60.2 ± 0.4 | 56.3 ± 0.6 |
> |  | **DIVER(ours)** | 45.6 ± 1.9 | 30.0 ± 0.0 | 83.9 ± 2.5 | 92.9 ± 0.1 | 50.6 ± 1.1 | 61.8 ± 0.7 | **60.8 ± 0.0** |
> | **k = 16** | Qwen2.5-Math-7B | 43.3 ± 12.0 | 23.3 ± 3.4 | 81.1 ± 3.0 | 89.9 ± 0.3 | 49.1 ± 0.8 | 56.5 ± 1.5 | 56.0 ± 2.2 |
> |  | GPRO w/ Clip-Higher | 45.6 ± 5.1 | 37.8 ± 5.1 | 84.3 ± 2.4 | 92.3 ± 0.3 | 59.3 ± 2.2 | 63.2 ± 0.5 | 63.7 ± 2.3 |
> |  | Entropy-RL | 47.8 ± 5.1 | 35.6 ± 2.0 | 88.0 ± 2.1 | 93.7 ± 0.2 | 50.7 ± 1.6 | 62.8 ± 0.3 | 63.2 ± 0.6 |
> |  | Pass@k Training | 50.0 ± 6.7 | 34.4 ± 8.4 | 82.7 ± 2.5 | 92.3 ± 0.1 | 50.6 ± 1.8 | 64.0 ± 0.5 | 62.3 ± 0.2 |
> |  | **DIVER(ours)** | 47.8 ± 5.1 | 38.9 ± 3.8 | 88.0 ± 0.0 | 94.4 ± 0.4 | 56.5 ± 2.9 | 66.0 ± 0.7 | **65.3 ± 1.0** |
> | **k = 32** | Qwen2.5-Math-7B | 52.2 ± 5.1 | 30.0 ± 3.3 | 88.8 ± 3.5 | 92.1 ± 0.5 | 54.4 ± 1.0 | 62.2 ± 0.8 | 63.3 ± 0.9 |
> |  | GPRO w/ Clip-Higher | 56.7 ± 10.0 | 42.2 ± 3.9 | 92.0 ± 2.8 | 94.4 ± 0.4 | 54.9 ± 0.4 | 65.8 ± 0.6 | 67.6 ± 0.8 |
> |  | Entropy-RL | 54.4 ± 2.0 | 42.2 ± 1.9 | 90.9 ± 3.0 | 94.1 ± 0.3 | 51.4 ± 0.3 | 66.3 ± 0.5 | 67.7 ± 0.6 |
> |  | Pass@k Training | 56.7 ± 5.8 | 33.3 ± 6.7 | 88.4 ± 0.7 | 93.8 ± 0.3 | 53.8 ± 1.5 | 68.0 ± 0.5 | 65.7 ± 1.7 |
> |  | **DIVER(ours)** | 58.9 ± 3.8 | 48.9 ± 1.9 | 92.4 ± 1.9 | 95.4 ± 0.2 | 64.4 ± 3.9 | 69.0 ± 0.3 | **70.5 ± 0.1** |
>
> [1] DAPO: An Open-Source LLM Reinforcement Learning System at Scale

---

> ### Author Response · Authors · 2025-11-21
> **Author Response (Part 3/3)**
>
> ***
>
> **`Q4. The diversity metrics (BLEU, formula overlap) may not effectively capture semantic-level reasoning diversity.`**
>
> A4. To verify whether TD and ED truly capture semantic-level reasoning diversity, we conduct a correlation analysis comparing them with two semantic similarity metrics: 1) **LLM-based scores**: We compute the cosine similarity of the model's hidden states from the final layer outputs using `Qwen2.5-7B-Instruct` for all baselines and our method; 2) **Embedding-based scores**: We use `google/embeddinggemma-300m` to encode responses into embeddings and compute their cosine similarity.
>
> For each method (DIVER and baselines), we generate multiple responses per prompt and compute all diversity metrics within each response group. Evaluated on 1,560 rollouts total, the average metrics are shown in the table (↑ indicates higher values represent greater diversity; ↓ indicates the opposite). The results demonstrate strong consistency between TD/ED and LLM-/embedding-based alternatives. **Methods achieving higher TD/ED scores consistently exhibit lower LLM-/embedding-based semantic similarity scores**, confirming that our choice of diversity metrics effectively captures meaningful semantic diversity rather than merely surface-level variation. **This validates the generality of TD and ED as reliable proxies for reasoning diversity**.
>
> | **Method** | **TD (↑)** | **ED (↑)** | **LLM-based (↓)** | **embedding-based (↓)** | pass@1 | pass@8 |
> | --- | --- | --- | --- | --- | --- | --- |
> | **DIVER(ours)** | 0.702 | 0.477 | 0.936 | 0.881 | 43.1 | 60.8 |
> | GRPO w/ Clip-Higher | 0.805 | 0.575 | 0.916 | 0.841 | 40.7 | 57.4 |
> | Entropy-RL | 0.549 | 0.294 | 0.947 | 0.954 | 41.8 | 58.0 |
> | Pass@k Training | 0.441 | 0.200 | 0.959 | 0.967 | 41.5 | 56.3 |
>
> Moreover, our diversity metrics are both computationally lightweight and easy to implement. **The computation overhead of calculating textual diversity and equational diversity is negligible**, since both metrics only involve rule-based calculations (e.g., n-gram matching for BLEU score or string recognition for equation extraction) without any feedforward or backpropagation of large-scale models (e.g., the above LLM-/embedding-based metrics).
>
> To validate whether our approach achieves higher quality and more meaningful reasoning results, we evaluate all test responses using `deepseek-ai/DeepSeek-V3.2-Exp`as a judge model, ranking responses generated by the base model, baselines, and DIVER for each prompt. The evaluation comprehensively assesses four key dimensions: **Completeness**, **Logical Consistency**, **Relevance**, and **Efficiency**. The following table presents the final result averaged over all six benchmarks: AIME24/25, AMC, MATH-500, Minerva, and Olympiad. It demonstrates DIVER's consistent superiority, achieving a 1.52 average ranking compared to baseline methods ranging from 2.57 to 3.19. DIVER solutions rank best (Rank 1) in 76.4% of cases—more than double the next best method. **This confirms our diversity measures capture meaningful improvements in reasoning quality**, not merely surface-level variations. We will include this insightful analysis on diversity metrics in the revised manuscript.
>
> | **Model** | **Avg. Rank(↓)** | **Rank 1(best)** | **Rank2** | **Rank 3** | **Rank4** | **Rank5 (worst)** |
> | --- | --- | --- | --- | --- | --- | --- |
> | Qwen2.5-Math-7B | 3.19 | 15.8% | 23.6%	 | 16.7%	 | 14.8%	 | 29.1% |
> | GPRO w/ Clip-Higher | 2.77	 | 21.1% | 23.9%	 | 24.9%	 | 17.8%	 | 13.2% |
> | Entropy-RL | 2.86	 | 22.9%	 | 19.7%	 | 22.0%	 | 22.9%	 | 12.5% |
> | Pass@k Training | 2.57	 | 31.7%	 | 22.0%	 | 16.8%	 | 17.7%	 | 11.8% |
> | **DIVER(ours)** | **1.52** | **76.4%**	 | 7.5%	 | 7.1%	 | 3.8%	 | 5.2% |

---

> ### Author Response · Authors · 2025-11-21
> **Author Summary Response**
>
> Thank you for your valuable review comments, which help us gain more critical insights and significantly enhance our empirical evaluation. **We are honored to have your positive comments on the motivation** (”the paper presents a clear motivation”), **the method** (a well-structured method )**, and the empirical evaluation with superior performance**  (”experiments are comprehensive and demonstrate consistent improvements”).
>
> Also, we are grateful that **other reviewers** (we refer to wris as R1, RcXm as R3, and BRzv as R4) **make positive comments on our overall contributions, including the motivation** (”studies an important problem” by R1, “the motivation…is clear and compelling” by R3, and “very simple, plug-and-play idea” by R4)**, the method** (“the method is simple and pricipled” by R3 and “easy to add to existing frameworks…the correctness mask and clipping are practical and effective” by R4)**, and the empirical evaluation with superior performance** (”comprehensive experimental results…DIVER achieves performance improvement across different benchmarks” by R1, “experimental evaluation is thorough” by R3, and “clear empirical gains…training diagnostics are helpful” by R4). We hope these collective evaluations could help clarify the breadth and strength of our contributions.
>
> **We have made a number of justifications and extended experimental analyses to address your concerns**. Due to limited time for posting the first round of rebuttal, **we will include more experimental results and update the paper thoroughly before the final discussion deadline Dec. 3rd**. Please let us know if we have addressed your concerns. We are more than delighted to have further discussions and improve our manuscript. If our response has addressed your concerns, we would be grateful if you could re-evaluate our work.

---

> > ### Comment · Reviewer_mNMY · 2025-11-23
> >
> > Thank you for your response. The additional clarifications and experiments have addressed most of my concerns. I suggest including these results in the final version of the paper. I have raised my score.

---

> > > ### Author Response · Authors · 2025-11-24
> > > **Thank you!**
> > >
> > > Thank you very much for your constructive feedback and your time in helping us improve our work. We sincerely appreciate your raising the score on our work! We will include these extensive results in our revised manuscript. We truly appreciate your continued engagement.

---

### Official Review · Reviewer_wris · 2025-11-03

**Soundness:** 3
**Presentation:** 3
**Contribution:** 2
**Rating:** 4
**Confidence:** 3

**Summary:**

This paper introduces DIVER (Diversity-Incentivized Exploration for Versatile Reasoning), a framework that uses global sequence-level diversity as an intrinsic reward to enhance exploration in reinforcement learning for Large Language Models (LLMs). The framework incorporates diversity incentives as an intrinsic reward and utilizes a potential-based reward shaping mechanism to ensure policy invariance and mitigate reward hacking. Empirical results demonstrate that DIVER outperforms existing RLVR baselines, improving performance on both in-domain and out-of-domain tasks, with notable improvements in the Pass@1 and Pass@k metrics.

**Strengths:**

1. This paper studies an important problem on how to boost the exploration

2. The paper provides comprehensive experimental results, evaluating DIVER on multiple benchmarks (in-domain and out-of-domain). It also assesses performance from different perspectives, including final benchmark performance and training dynamics.

3. DIVER achieves performance improvement across different benchmarks, but the improvement is not consistent compared to baselines methods.

**Weaknesses:**

1. While Textual Diversity (TD) and Equational Diversity (ED) are designed to incentivize exploration, it is unclear whether they truly measure meaningful diversity in reasoning. For example, ED could be easily hacked by generating irrelavent formulas. As shown in figure 4, DIVER leads to a decrease in the solve_all rate, which raises concerns about whether the model is exploring paths of lower quality.


2. The paper claims that "Longer Horizons Improve Performance", but as shown in Figure 7 and 8, all experiments use response length at 1-2k tokens. It remains unclear whether this finding holds for thinking models with longer horizon reasonings. Verification on such models is needed.

3. More comprehensive ablations are necessary to demonstrate the effectiveness of the proposed method. As show in Figure 9, a larger $\lambda$ could bring almost no performance improvement.

**Questions:**

For rollout data that already exhibits high entropy, is it still necessary to introduce additional diversity incentives? Could this approach unintentionally encourage false positives, where the model generates diverse but incorrect responses? How does DIVER ensure that diversity incentives do not lead to an over-exploration of irrelevant or incorrect reasoning paths?

---

> ### Author Response · Authors · 2025-11-21
> **Author Response (Part 1/3)**
>
> ***
>
> **`Q1. While Textual Diversity (TD) and Equational Diversity (ED) are designed to incentivize exploration, it is unclear whether they truly measure meaningful diversity in reasoning.`**
>
> A1. To verify whether TD and ED truly capture semantic-level reasoning diversity, we conduct a correlation analysis comparing them with two semantic similarity metrics: 1) **LLM-based scores**: We compute the cosine similarity of the model's hidden states from the final layer outputs using `Qwen2.5-7B-Instruct` for all baselines and our method; 2) **Embedding-based scores**: We use `google/embeddinggemma-300m` to encode responses into embeddings and compute their cosine similarity.
>
> For each method (DIVER and baselines), we generate multiple responses per prompt and compute all diversity metrics within each response group. Evaluated on 1,560 rollouts total, the average metrics are shown in the table (↑ indicates higher values represent greater diversity; ↓ indicates the opposite). The results demonstrate strong consistency between TD/ED and LLM-/embedding-based alternatives. **Methods achieving higher TD/ED scores consistently exhibit lower LLM-/embedding-based semantic similarity scores**, confirming that our choice of diversity metrics effectively captures meaningful semantic diversity rather than merely surface-level variation. **This validates the generality of TD and ED as reliable proxies for reasoning diversity**.
>
> | **Method** | **TD (↑)** | **ED (↑)** | **LLM-based (↓)** | **embedding-based (↓)** | pass@1 | pass@8 |
> | --- | --- | --- | --- | --- | --- | --- |
> | **DIVER(ours)** | 0.702 | 0.477 | 0.936 | 0.881 | 43.1 | 60.8 |
> | GRPO w/ Clip-Higher | 0.805 | 0.575 | 0.916 | 0.841 | 40.7 | 57.4 |
> | Entropy-RL | 0.549 | 0.294 | 0.947 | 0.954 | 41.8 | 58.0 |
> | Pass@k Training | 0.441 | 0.200 | 0.959 | 0.967 | 41.5 | 56.3 |
>
> Moreover, our diversity metrics are both computationally lightweight and easy to implement. **The computation overhead of calculating textual diversity and equational diversity is negligible**, since both metrics only involve rule-based calculations (e.g., n-gram matching for BLEU score or string recognition for equation extraction) without any feedforward or backpropagation of large-scale models (e.g., the above LLM-/embedding-based metrics).
>
> To validate whether our approach achieves higher quality and more meaningful reasoning results, we evaluate all test responses using `deepseek-ai/DeepSeek-V3.2-Exp`as a judge model, ranking responses generated by the base model, baselines, and DIVER for each prompt. The evaluation comprehensively assesses four key dimensions: **Completeness**, **Logical Consistency**, **Relevance**, and **Efficiency**. The following table presents the final result averaged over all six benchmarks: AIME24/25, AMC, MATH-500, Minerva, and Olympiad. It demonstrates DIVER's consistent superiority, achieving a 1.52 average ranking compared to baseline methods ranging from 2.57 to 3.19. DIVER solutions rank best (Rank 1) in 76.4% of cases—more than double the next best method. **This confirms our diversity measures capture meaningful improvements in reasoning quality**, not merely surface-level variations. We will include this insightful analysis on diversity metrics in the revised manuscript.
>
> | **Model** | **Avg. Rank(↓)** | **Rank 1(best)** | **Rank2** | **Rank 3** | **Rank4** | **Rank5 (worst)** |
> | --- | --- | --- | --- | --- | --- | --- |
> | Qwen2.5-Math-7B | 3.19 | 15.8% | 23.6%	 | 16.7%	 | 14.8%	 | 29.1% |
> | GPRO w/ Clip-Higher | 2.77	 | 21.1% | 23.9%	 | 24.9%	 | 17.8%	 | 13.2% |
> | Entropy-RL | 2.86	 | 22.9%	 | 19.7%	 | 22.0%	 | 22.9%	 | 12.5% |
> | Pass@k Training | 2.57	 | 31.7%	 | 22.0%	 | 16.8%	 | 17.7%	 | 11.8% |
> | **DIVER(ours)** | **1.52** | **76.4%**	 | 7.5%	 | 7.1%	 | 3.8%	 | 5.2% |
>
> ***
>
> **`Q2. For example, ED could be easily hacked by generating irrelevant formulas.`**
>
> A2. As detailed in our A6 to Q6, we adopt **balanced shaping** and **conditional shaping** mechanisms to mitigate reward hacking. By clipping the diversity reward and applying it exclusively to correct answers, we prevent the model from exploiting the reward by generating irrelevant formulas. Furthermore, Appendix D: Case Study (L1026, P20) demonstrates that **DIVER generates diverse yet logically coherent formulas across multiple responses**, confirming that DIVER’s deep exploration is mathematically meaningful rather than arbitrary or irrelevant.

---

> ### Author Response · Authors · 2025-11-21
> **Author Response (Part 2/3)**
>
> ***
>
> **`Q3. As shown in figure 4, DIVER leads to a decrease in the solve_all rate, which raises concerns about whether the model is exploring paths of lower quality.`**
>
> A3. As shown in Figure 4 (L378, P8), while DIVER shows a decrease in solve_all rate compared to baselines, it achieves the lowest solve_none rate that consistently decreases throughout training. This is consistent with our preliminary analysis in Sec. 3.2 (L185, P4), where high-diversity training simultaneously yields a lower solve_all rate (1.1 points decrease) and a lower solve_none rate (3.1 points decrease). Intuitively, the high-diversity scheme trades off a small fraction of responses to explore a wider range of reasoning patterns, enabling the search for novel breakthroughs when conventional solutions fail. This property is particularly appealing in practice, since **achieving success on complex problems carries greater value than obtaining uniformly correct answers on simple questions**.
>
> Also, a prompt with all accurate or all wrong responses will produce a zero advantage and zero policy gradient, reducing training efficiency. A lower solve_all or solve_none rate can benefit GRPO training, providing more useful information for exploration. By maintaining high diversity during training, DIVER reduces the proportion of such ineffective prompts (solve_all and solve_none), thereby preventing premature convergence to local optima. As demonstrated in Table 1 (L325, P7), DIVER achieves superior performance on both in-domain and out-of-domain benchmarks compared to baselines, confirming that **it successfully explores higher-quality solution paths despite the temporary trade-off in solve_all rate**.
>
> ***
>
> **`Q4. The paper claims that "Longer Horizons Improve Performance", but as shown in Figure 7 and 8, all experiments use response length at 1-2k tokens. It remains unclear whether this finding holds for thinking models with longer horizon reasonings. Verification on such models is needed.`**
>
> A4. **The horizon in this claim (Sec. 4.4, L464, P9) refers to the window size used for calculating the diversity metrics, rather than the response length**. For instance, a horizon of 200 indicates that we compute Textual Diversity (TD) and Equational Diversity (ED) metrics only on the first 200 tokens of a generated response. The complete response length can be significantly longer, as shown in the Response Length graph in Figure 8, where full responses reach 1600-1800 tokens. We appreciate this careful observation and will clarify this distinction in the revised paper.
>
> Following your suggestion, we have conducted new experiments to evaluate DIVER and baselines, using DeepSeek-R1-Distill-Qwen-7B as the base model that exhibits a longer reasoning horizon (full responses reach 2500-3500 tokens). The following result demonstrates **DIVER’s consistent superiority in reasoning capacity on models with longer reasoning horizons**.
>
> | **Method** | **In-Domain** |  |  |  |  |  | **Out-of-Domain** |  |  |  |
> | --- | --- | --- | --- | --- | --- | --- | --- | --- | --- | --- |
> |  | **AIME24/25** | **AMC** | **MATH-500** | **Minerva** | **Olympiad** | **Avg.** | **ARC-c** | **GPQA*** | **MMLU-Pro** | **Avg.** |
> | **DeepSeek-R1-Distill-Qwen-7B** | 40.6/30.0 | 72.3 | 85.8 | 28.3 | 51.1 | 51.4 | 27.4 | 0.0 | 45.8 | 24.4 |
> | **GPRO w/ Clip-Higher** | 49.3/35.0 | 81.7 | 90.2 | 35.7 | 59.0 | 58.5 | 85.2 | 2.6 | 54.9 | 47.5 |
> | **DVIER(ours)** | 51.1/36.9 | 82.0 | 92.6 | 36.8 | 61.2 | **60.1** | 85.6 | 10.7 | 56.6 | **50.9** |

---

> ### Author Response · Authors · 2025-11-21
> **Author Response (Part 3/3)**
>
> ***
>
> **`Q5. More comprehensive ablations are necessary to demonstrate the effectiveness of the proposed method. As shown in Figure 9, a larger $\lambda$ could bring almost no performance improvement.`**
>
> A5. The reward shaping coefficient $\lambda$ controls the strength of diversity incentives in our method as $R' = R + \lambda\cdot R_{\text{int}}$. As Figure 9 demonstrates, **DIVER exhibits remarkable robustness across different $\lambda$ values**. Surprisingly, even a small coefficient ($\lambda=0.001$) can provide sufficient diversity incentive. The possible reason is that, as the original reward $R$ is binary, a small scale of the intrinsic reward $R_{\text{int}}$ can already influence the landscape of the reward distribution. Then, increasing the value ($\lambda=0.01/0.1$) shows only minor performance variations without triggering reward hacking behaviors. Finally, with a large value $\lambda=1$, DIVER’s performance degrades a little. When $\lambda = 0$, the method reduces to baseline GRPO, as shown in Table 1 (L325, P7). **This hyperparameter analysis demonstrates the robustness of our reward shaping mechanism, making DIVER practical for applications without requiring precise tuning.**
>
> As DIVER’s core is to formulate global diversity as an intrinsic reward to incentivize deep exploration, **we conducted the ablation analysis on core components,** such as the coefficient  $\lambda$ (Appendix B.2, L836, P16) and reward hacking solutions (Sec. 4.4, L450, P9). Due to the limited time for posting the first round of rebuttal, we did not have enough time to conduct more ablation studies to analyze the impact of other trivial hyperparameters, such as the group size and temperature. We will continue to conduct more ablation studies to further demonstrate DIVER’s effectiveness, and upload the results before the final discussion deadline Dec. 3rd.
>
> ***
>
> **`Q6. For rollout data that already exhibits high entropy, is it still necessary to introduce additional diversity incentives? Could this approach unintentionally encourage false positives, where the model generates diverse but incorrect responses? How does DIVER ensure that diversity incentives do not lead to an over-exploration of irrelevant or incorrect reasoning paths?`**
>
> A6.  We agree that naively encouraging diversity can increase the risk of reward hacking, where the model may over-explore diverse but incorrect responses to obtain high intrinsic rewards. As shown in Sec. 3.5 (L294, P6), **we have designed simple and easy-to-use mechanisms to mitigate this risk**.
>
> First, we adopt **balanced shaping** by clipping the diversity reward as  $r_i^\text{int} = \text{clip}(r_i^\text{int}; 0, \sigma)$, where $\sigma$ serves as a predetermined upper bound that prevents the model from excessively exploiting the shaping reward. It ensures that diversity doesn't come at the expense of solution accuracy. Second, we implement **conditional shaping**, which applies diversity rewards exclusively to correct responses through the formula $r_i' = r_i + \lambda \cdot r_\text{int} \cdot I(r_i)$, where $I(r_i)$ equals 1 only for correct solutions and 0 otherwise. This ensures diversity incentives exclusively reward genuinely correct solutions, preventing the model from trading off correctness for diversity. As shown in Figure 7 (L450, P9), **this mechanism effectively mitigates reward hacking concerns**.

---

> ### Author Response · Authors · 2025-11-21
> **Author Summary Response**
>
> Thank you for your valuable review comments, which help us gain more critical insights and significantly enhance our empirical evaluation. **We are honored to have your positive comments on the technical challenge** (”studies an important problem on how to boost the exploration”), **the empirical evaluation** (”provides comprehensive experimental results”), **and our superior performance** (”DIVER achieves performance improvement across different benchmarks”).
>
> Also, we are grateful that **other reviewers** (we refer to mNMY as R2, RcXm as R3, and BRzv as R4) **make positive comments on our overall contributions, including the motivation** (”presents a clear motivation” by R2, “the motivation…is clear and compelling” by R3, and “very simple, plug-and-play idea” by R4)**, the method** (”a well-structured method” by R2, “the method is simple and pricipled” by R3, and “easy to add to existing frameworks…the correctness mask and clipping are practical and effective” by R4)**, and the empirical evaluation with superior performance** (”experiments are comprehensive and demonstrate consistent improvements” by R2, “experimental evaluation is thorough” by R3, and “clear empirical gains…training diagnostics are helpful” by R4). We hope these collective evaluations could help clarify the breadth and strength of our contributions.
>
> **We have made a number of justifications and extended experimental analyses to address your concerns**. Due to limited time for posting the first round of rebuttal, **we will include more experimental results and update the paper thoroughly before the final discussion deadline Dec. 3rd**. Please let us know if we have addressed your concerns. We are more than delighted to have further discussions and improve our manuscript. If our response has addressed your concerns, we would be grateful if you could re-evaluate our work.

---

### Author Response · Authors · 2025-12-01
**Author Rebuttal Summary**

Dear ACs, SACs, and PCs,

We want to thank all the reviewers (we refer to wris as R1, mNMY as R2, RcXm as R3, and BRzv as R4) for their comprehensive comments to improve our work. **When releasing the initial review comments, we were honored to have a lot of positive feedback on our overall contributions, including the motivation** (”studies an important problem” by R1, ”the paper presents a clear motivation” by R2, “the motivation…is clear and compelling” by R3, and “very simple, plug-and-play idea” by R4), **the method** (”a well-structured method” by R2, “the method is simple and principled” by R3 and “easy to add to existing frameworks…the correctness mask and clipping are practical and effective” by R4), **and the empirical evaluation with superior performance** (”comprehensive experimental results…DIVER achieves performance improvement across different benchmarks” by R1, ”experiments are comprehensive and demonstrate consistent improvements” by R2, “experimental evaluation is thorough” by R3, and “clear empirical gains…training diagnostics are helpful” by R4).

Before the information leakage on Nov. 27th, we had already addressed the primary concerns raised by the reviews:

- For R1, we validated that our diversity metrics truly capture meaningful reasoning diversity rather than surface-level variations, demonstrated that our approach generates high-quality reasoning paths verified by a judge model, and provided additional experiments on models with longer reasoning horizons.
- For R2, we extended our evaluation to larger values of $k$ (up to 1024), implemented a fair two-stage training comparison with Pass@k Training, and provided statistical significance through multiple runs with means and standard deviations reported. **R2 explicitly acknowledged that our "additional clarifications and experiments have addressed most of my concerns", and raised his/her score from 4 to 6 on 15:18, Nov. 23rd, 2025.**
- For R3, we demonstrated the generality of our diversity metrics by correlating them with Embedding-based and LLM-based measures (which represent semantic-level diversity), quantified the minimal computational overhead (only 5% increase), and implemented their suggestion to combine diversity metrics, which yielded even better performance.
- For R4, we clarified the consistency between our theoretical reward-shaping framework and practical implementation, detailed the equation extraction process with accuracy metrics, provided comprehensive ablation studies on key hyperparameters, and quantified the minimal computational overhead.

All these responses have been meticulously documented in our rebuttal, and we have committed to incorporating these extended results and clarifications in our revised manuscript. We believe that our paper offers valuable contributions to the field, as evidenced by the initial positive feedback and the improved rating after our rebuttal.

Best regards,

The Authors

---

> ### Author Response · Authors · 2025-12-03
> **Detailed Summary of Reviewer Concerns and Response Status**
>
> ## Reviewer wris Score 4 Conf 3 (No Reply)
>
> > While TD and ED are designed to incentivize exploration, it is unclear whether they truly measure meaningful diversity in reasoning.
> >
>
> We validated our metrics against LLM-based and embedding-based semantic similarity measures (**Table 7 of Appendix D.1**). Using DeepSeek-V3.2-Exp as judge, DIVER solutions ranked best in 76.4% of cases (**Figure 9 of Section 4.4; Table 8 of Appendix D.2**).
>
> > ED could be easily hacked by generating irrelevant formulas.
> >
>
> Our balanced and conditional shaping mechanisms mitigate reward hacking by applying diversity rewards exclusively to correct answers (**Section 3.5; Figure 7 of Section 4.4**).
>
> > DIVER leads to a decrease in the solve_all rate, raising concerns about exploring paths of lower quality.
> >
>
> While solve_all rate decreases slightly, DIVER achieves the lowest solve_none rate, exploring wider reasoning patterns that enable novel breakthroughs when conventional solutions fail (**Section 3.2**).
>
> > Does DIVER remain effective for longer reasoning sequences or different models?
> >
>
> We validated DIVER with DeepSeek-R1-Distill-Qwen-7B (generating 2500-3500 token responses), outperforming GPRO w/ Clip-Higher by +1.6 and +3.4 points on in-domain and out-of-domain tasks (**Table 9 of Appendix D.3**).
>
> ## Reviewer mNMY Score **4→6** Conf 4 (Reply in 23 Nov)
>
> > The improvement on Pass@k is relatively small and only evaluated up to k = 32.
> >
>
> We extended experiments to k=128, 256, 512, and 1024, showing DIVER consistently outperforms all baselines, reaching 86.7% on AIME24 and 100% on AMC at Pass@1024 (**Table 2 of Section 4.3**).
>
> > The comparison on Pass@1 in Table 1 seems unfair, since methods like Pass@k Training mainly aim to improve Pass@k.
> >
>
> We re-ran experiments with two-stage training (optimizing Pass@k, then fine-tuning on Pass@1). DIVER still consistently outperforms Pass@k Training across both domains (**Table 1 of Section 4.2**).
>
> > GRPO remains the best baseline, which seems inconsistent with expected performance.
> >
>
> Our GRPO baseline uses the clip-higher technique from DAPO. Repeated experiments across Pass@2 to Pass@32 validate DIVER's consistent superiority (**Table 5 of Appendix C.1**).
>
> > Do these metrics actually capture semantic-level reasoning diversity?
> >
>
> To verify our metrics capture semantic-level reasoning diversity, we correlated TD/ED with LLM-based and embedding-based similarity metrics (**Table 7 of Appendix D.1**).
>
> ## Reviewer RcXm Score 6 Conf 4 (No Reply)
>
> > Limited generality of diversity metrics: The choice of diversity metrics lacks clear guidance.
> >
>
> Following the suggestion, we implemented DIVER-MIX combining TD and ED, achieving better performance than either metric alone (**Table 1 of Section 4.2**). Metrics implementation details are in **Appendix C.2 and C.3**.
>
> > No analysis of computational cost.
> >
>
> DIVER incurs only 5% increased training time vs. GRPO/Entropy-RL baselines (**Table 3 of Appendix B.2**).
>
> > Insufficient ablations and sensitivity analysis of key components and hyperparameters.
> >
>
> DIVER exhibits robustness across different λ values (**Figure 10 of Appendix D.4**), with additional analyses for group size and temperature confirming consistent performance (**Table 10 of Appendix D.4**).  And  conditional shaping mechanism prevents reward hacking while ensuring effective global exploration (**Figures 7, 8 of Section 4.4, Figure 11 of Appendix D.4**).
>
> ## Reviewer BRzv Score 6 Conf 2 (No Reply)
>
> > Theory vs practice mismatch: The shaping argument assumes state-only potential while the bonus is group-dependent.
> >
>
> Our reward-shaping design works by setting the shaping reward as the difference between diversities of adjacent states, deriving the intrinsic reward as the diversity of the final response (**Appendix A**).
>
> > Equational diversity extraction is under-specified.
> >
>
> We extract mathematical formulas using regular expressions with ~8.4% failure rate, which is sufficient for guiding effective exploration (**Section 3.4 and Appendix C.3**).
>
> > Training overhead concerns vs GRPO?
> >
>
> DIVER incurs only 5% increased training time, with negligible computation overhead for diversity calculations (**Table 3 of Appendix B.2**).
>
> > Insufficient ablations and sensitivity analysis of key hyperparameters.
> >
>
> DIVER exhibits robustness across different λ values (**Figure 10 of Appendix D.4**), with additional analyses for group size and temperature confirming consistent performance (**Table 10 of Appendix D.4**).

---

### Meta-Review · Area_Chair_x174 · 2026-01-01

**Summary:**

This paper proposes DIVER, a reinforcement learning framework for large language model reasoning that introduces global sequence-level diversity as an intrinsic reward under the RL with verifiable rewards paradigm. The core idea is to explicitly incentivize exploration in the vast and sparse reasoning space by rewarding diversity at the sentence and equation level, implemented through textual diversity and equational diversity metrics. These intrinsic rewards are incorporated via a potential-based reward shaping mechanism to preserve policy invariance, along with practical techniques such as clipping and correctness-conditioned shaping to mitigate reward hacking. Extensive experiments across in-domain and out-of-domain mathematical reasoning benchmarks demonstrate consistent improvements over strong baselines such as GRPO with clipping, Entropy-RL, and Pass@k Training, across both Pass@1 and Pass@k (up to k=1024), as well as robustness to longer reasoning horizons.

The primary strengths of the paper are its clear and well-motivated focus on exploration deficiencies in RLVR, the conceptual simplicity and plug-and-play nature of the proposed diversity incentives, and the breadth and depth of empirical evaluation. The rebuttal substantially strengthened the work by validating that the proposed diversity metrics correlate with semantic-level diversity, extending Pass@k evaluations to very large k, clarifying fairness in comparisons via two-stage training, quantifying computational overhead, and demonstrating effectiveness on longer-horizon reasoning models. Reviewers’ post-rebuttal comments generally acknowledged that the major technical and empirical concerns had been addressed. The remaining weaknesses are relatively minor: the diversity metrics remain heuristic in nature, and some ablation dimensions (e.g., secondary hyperparameters) could still be expanded; however, these limitations are clearly acknowledged and do not undermine the main conclusions.

In light of the strengthened empirical evidence and satisfactory resolution of the reviewers’ core concerns, this paper should be accepted.

**Reviewer Concerns:**

The rebuttal convincingly addressed the main technical and empirical criticisms raised by all reviewers. Questions about whether textual and equational diversity capture meaningful reasoning diversity were answered through correlation analyses with embedding-based and LLM-based semantic similarity metrics, as well as judge-model evaluations demonstrating higher-quality reasoning outputs. Concerns about reward hacking and degraded solution quality were mitigated by clear explanations and additional experiments showing the effectiveness of clipping and correctness-conditioned shaping. Issues regarding limited Pass@k evaluation and fairness of Pass@1 comparisons were resolved by extending experiments to k=1024 and by re-running Pass@k Training with a two-stage protocol. The authors also addressed concerns about generality and computational cost by reporting minimal overhead and by validating performance on longer-horizon reasoning models.

The remaining concerns are relatively minor and largely about scope rather than correctness. While the authors provided additional ablations and sensitivity analyses, some reviewers may still prefer even broader exploration of secondary hyperparameters or alternative diversity formulations. Additionally, although the diversity metrics are empirically validated, they remain heuristic proxies rather than theoretically grounded measures of semantic reasoning diversity. These issues, however, are not critical blockers and do not substantially detract from the paper’s contributions.

**Reviewer Scores:**

Based on the rebuttal and discussion, Reviewer R2 explicitly increased their score from 4 to 6 and indicated that most concerns were resolved. Reviewers R3 and R4, who initially gave scores of 6, would likely maintain or slightly strengthen their positive assessments given the added experiments, clarifications, and robustness analyses. Reviewer R1, who initially scored the paper as marginally below threshold, would likely revise their score upward (e.g., from 4 to around 6) given that their primary concerns about semantic diversity, reward hacking, and longer-horizon reasoning were directly addressed with new evidence.

---

### Decision · Program_Chairs · 2026-01-26

Accept (Poster)